# HIERARCHICAL ROUTERS FOR EFFICIENT TOP-k RETRIEVAL IN SPARSE ATTENTION

## ABSTRACT

Attention mechanisms have achieved remarkable success in deep learning through parallel searching for the most relevant tokens in large-scale data. However, both the memory and computational costs of self-attention scale quadratically with sequence length, making it infeasible for long sequences. Recent sparse top-$k$ attention methods can achieve performance comparable to full attention with much lower memory and computational overhead. Nevertheless, they often rely on graph- or tree-based index structures, which are too slow for batches of token sequences to rebuild across layers or heads, or use partition-based techniques which lack precision. To address this issue, we propose a search algorithm for sparse attention: Hierarchical Router Algorithm, HIROUTER, which can efficiently construct indexing structures and dynamically retrieve top-k tokens on a per-sequence basis, striking a better balance between speed and accuracy. HIROUTER employs a multi-level routing mechanism that hierarchically partitions tokens into discrete buckets along a learned tree structure with $\mathcal{O}(T)$ to the sequence length $T$. Notably, our dual entropy loss directly regularizes embeddings, using affinity for stronger sample–centroid alignment to improve top-$k$ recall and balanced buckets to ensure efficient GPU parallelism. HIROUTER outperforms FlashAttention in speed on long sequences while matching or surpassing the accuracy of full attention, offering a compelling solution for scalable and efficient attention mechanisms.

## 1 INTRODUCTION

Transformers (Vaswani et al., 2017) have become indispensable for sequence modeling across a wide range of domains (OpenAI, 2023), including natural language processing (NLP) (Devlin et al., 2019; Brown et al., 2020; OpenAI, 2023; Jiang et al., 2024), computer vision (Dosovitskiy et al., 2021; Ramesh et al., 2021; Brooks et al., 2024), and more. At the heart of Transformer models lies the self-attention mechanism (Vaswani et al., 2017), which constructs rich token representations by attending to all elements in a sequence in parallel. This innovation has powered breakthroughs in language modeling (Radford et al., 2019), machine translation (Ott et al., 2018), text generation (Brown et al., 2020), image classification (Touvron et al., 2021), video generation (Brooks et al., 2024), and beyond. Despite its success, self-attention incurs $\mathcal{O}(T^2)$ memory and computational costs as the sequence length $T$ grows, posing a major obstacle for long-sequence applications (Child et al., 2019; Beltagy et al., 2020; Tay et al., 2021). This quadratic cost often makes naive self-attention prohibitively expensive for real-world, large-scale applications.

Recent research has proposed several strategies to mitigate the complexity of self-attention. Deng et al. (2024) show that attention matrices are inherently sparse. Building on this observation, **Top-$k$ attention** restricts computation to the $k$ most informative tokens, substantially reducing both memory usage and FLOPs while maintaining full-attention quality (Roy et al., 2021; Kitaev et al., 2020; Gupta et al., 2021; Bertsch et al., 2023; Mao et al., 2024). Nevertheless, existing top-$k$ implementations suffer from two fundamental drawbacks: **(i) Inefficient time–precision trade-off**, as they rely on generic $k$-Nearest Neighbors ($k$-NN) or Maximum Inner-Product Search (MIPS) routines that are ill-suited for batches of token sequences in attention. Consequently, these methods rebuild graph- or tree-based indices for each head and layer, inserting tokens sequentially and forgoing GPU parallelism, which leads to prohibitive runtime (Kitaev et al., 2020; Roy et al., 2021); **(ii) Inefficient GPU utilization**, as their dependance on data-agnostic $k$-NN libraries (Johnson et al., 2021; Guo et al., 2020) that ignore the underlying token distribution (Johnson et al., 2021; Malkov

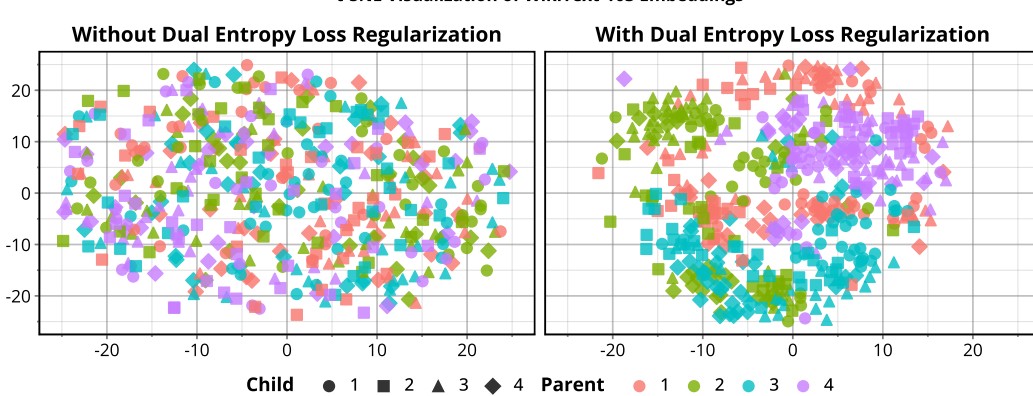

Figure 1: Illustration of HIROUTER configured with two levels, each containing four buckets. Colors denote **parent** buckets (level 1) and marker shapes denote **child** buckets (level 2). With regular attention (left), embeddings are scattered across buckets. In HIROUTER (right), embeddings with the same shape and color cluster tightly, ensuring that semantically similar tokens fall into the same bucket. This improves top-$k$ retrieval, as queries need only search within well-formed buckets instead of competing with irrelevant tokens.

& Yashunin, 2020; Guo et al., 2020), and therefore fail to leverage neural networks' ability to learn data-aware indices. Partition-based methods such as LSH (Kitaev et al., 2020) or $k$-means (Roy et al., 2021) further exacerbate this issue by producing imbalanced buckets under skewed data distributions, leading to inefficient GPU occupancy.

In this work, we address both limitations through our hierarchical routing algorithm, HIROUTER, together with a dual-entropy loss regularizer: **(i) striking a good balance between speed and precision** by routing tokens *in parallel* into a multi-level tree and performing high-recall MIPS entirely on-GPU, per sequence and per head, making the approach well suited to batched long-sequence data. Importantly, the dual-entropy loss regularizes similar tokens to cluster together, thereby improving retrieval precision, as illustrated in Figure 1. **(ii) improving GPU utilization** by partitioning KV-cache into *equal-sized* buckets. Specifically, the dual-entropy loss encourages the embeddings to form uniform partitions, and the Gumbel–Softmax relaxation makes the discrete routing differentiable, allowing the entire partitioning scheme to be optimized end-to-end.

Our experiments in a wide range of tasks on the evaluation benchmarks for language modeling, natural language understanding show that hierarchical routers achieve better performance compared to strong transformer baselines. Extensive evaluations demonstrate that our method achieves substantial improvements in computational efficiency and retrieval accuracy over existing top-$k$ retrieval methods. We summarize our key contributions as follows:

I) **Efficient Parallel Hierarchical Top-$k$ Attention:** We introduce HIROUTER, a hierarchical router that clusters tokens in parallel into multi-level buckets, enabling top-$k$ retrieval with complexity $\mathcal{O}(kT), k \ll T$ while outperforming or matching full-attention performance.

II) **Entropy-Based Dual-Objective Regularization:** We propose a routing loss that regularizes key and query embeddings, balancing bucket loads while tightening token–centroid affinity. This ensures retrieval quality and balanced, equal-sized buckets for higher GPU utilization.

III) **Strong Benchmarks and Efficient Scalability** On language modeling and reasoning benchmarks, HIROUTER achieves strong accuracy with efficiency, balancing performance and cost. It also provides up to $3.55\times$ speedup on long-context inputs over FlashAttention.

## 2 RELATED WORK

**Efficient Attention.** Location-based sparse patterns have long been used to curb the quadratic complexity of vanilla self-attention. Early works alternated coarse and local windows to reduce the receptive field (Liu et al., 2018). Strided/dilated patterns were later adopted for image generation (Child

et al., 2019), while adaptive windows offered dynamic sparsity for sequence modeling (Sukhbaatar et al., 2019). Global-plus-local hybrids such as Longformer (Beltagy et al., 2020), ETC (Ainslie et al., 2020), and BigBird (Zaheer et al., 2020) designate a small set of global tokens that attend everywhere. Orthogonal to fixed patterns, low-rank or kernelized approaches approximate dense attention via linear projections (Katharopoulos et al., 2020; Xiong et al., 2021; Wang et al., 2020) or random features (Choromanski et al., 2021; Peng et al., 2021). NSA (Yuan et al., 2025) is a natively trainable sparse attention that combines hierarchical token compression and selection. While these designs bring linear or near-linear complexity, they often under-use fine-grained, content-based interactions.

**Sparse Top-K Attention.** Content-based sparsification keeps only the most relevant tokens per query (Pagliardini et al., 2023). Routing Transformers (Roy et al., 2021) and Reformer (Kitaev et al., 2020) hash queries and keys into shared buckets. Memory-Efficient Top-$k$ Attention (Gupta et al., 2021) and Unlimiformer (Bertsch et al., 2023) push context lengths toward millions of tokens, but they still depend on external $k$-NN or hashing modules. IceFormer (Mao et al., 2024) improves transformer efficiency by integrating ANN search mechanism that focuses on the $k$-NN results as the most relevant tokens during inference, bypassing the need to compute the full attention matrix. ZETA (Zeng et al., 2025) proposes using Z-order curve projections to enable efficient parallel top-k token retrieval in long-sequence self-attention. Haris (2025) provide theoretical sampling guarantees and gradient estimation techniques that complement our architectural implementation. In contrast, Chalkidis et al. (2022) proposes HAT, which utilizes a fixed, segment-based topology optimized specifically for long-document classification. However, most existing pipelines either incur substantial overhead by constructing exact indices for each head or tolerate a significant drop in recall when relying on approximate hashing.

**Approximate Top-$k$ Retrieval.** Classical similarity search relies on graph or tree indices such as HNSW (Malkov & Yashunin, 2020), IVFPQ in FAISS (Douze et al., 2024), or ScaNN (Guo et al., 2020), which build indices sequentially and are ill-suited for per-layer GPU parallelism in deep Transformers. Inspired by learned-index approaches (Kraska et al., 2018; Li et al., 2023; Gupta et al., 2022), we instead propose a learnable hierarchical router that jointly trains centroids and routing logits, removes explicit indexing overhead, and adapts dynamically to data. Unlike offline-trained partitioners (Dong et al., 2023), HIROUTER updates embeddings of keys, queries, and centroids on-the-fly with entropy regularization, improves the precision of top-$k$ retrieval, and balances buckets.

**KV Cache Compression.** Recent KV-cache management methods have also introduced hierarchy-based clustering approaches (Fu et al., 2025; Wang et al., 2025; Liu et al., 2025; Tu et al., 2025; Zandieh et al., 2024), with the objective of optimizing inference-time storage efficiency via budgeting, semantic clustering, or sublinear sampling. In contrast, HIROUTER optimizes the *retrieval* process itself, addressing how attention is computed rather than how the cache is stored or pruned.

## 3 METHODOLOGY

In this section, we present HIROUTER, a hierarchical routing framework for efficient top-$k$ attention. After reviewing self-attention and top-$k$ variants in Section 3.1, we introduce top-$k$ retrieval algorithms, HIROUTER, in Section 3.2 with entropy-based regularization that simultaneously sharpens token–centroid alignment and balances bucket occupancy in Section 3.4. Finally, a hierarchical beam search (Section 3.5) retrieves the candidate buckets for sparse attention.

### 3.1 PRELIMINARIES

**Self-Attention.** Self-attention (Vaswani et al., 2017) lies at the heart of modern sequence models, enabling each token to attend to all others and thereby capture long-range dependencies. Given queries $\boldsymbol{Q} \in \mathbb{R}^{T \times d}$, keys $\boldsymbol{K} \in \mathbb{R}^{T \times d}$, and values $\boldsymbol{V} \in \mathbb{R}^{T \times d}$, the scaled dot-product attention is

$$\text{Attention}(\boldsymbol{Q}, \boldsymbol{K}, \boldsymbol{V}) = \text{softmax}\left(\boldsymbol{Q}\boldsymbol{K}^\top / \sqrt{d}\right) \boldsymbol{V}, \tag{1}$$

where $d$ denotes the common dimensionality of queries, keys, and values.

---

**Algorithm 1** HiRouter Pseudocode

---

**Require:** Keys $\boldsymbol{K}$, Values $\boldsymbol{V}$, Query $\boldsymbol{q}$. Router centroids $\mathcal{C}_{\text{all}}^{(l)}$ at each level $l \in \{1, \dots, L\}$, beam width $M$.
   **Output:** Attention scores $\boldsymbol{A}$
 1: **Initialize:**
 2:   Accumulated probabilities $\boldsymbol{P}_K^0 = \boldsymbol{1}$, $\boldsymbol{P}_q^0 = \boldsymbol{1}$.
 3:   Root indices for keys and query:
 4:     $\boldsymbol{a}_K^0 = \boldsymbol{0}$ // each key starts at root bucket 0
 5:     $\boldsymbol{a}_q^0 = [0, \dots, 0] \in \mathbb{R}^M$ // query beam: $M$ candidates, all at root
 6: **for** level $l \in \{1, \dots, L\}$ **do**
 7:     **1. Gather active child centroids:**
 8:       // at level $l$, we only consider child centroids of the current parents
 9:       // for keys: parents $\boldsymbol{a}_K^{l-1}$; for query: beam parents $\boldsymbol{a}_q^{l-1}$

$$\mathcal{C}_K^{(l)} = \text{GatherChildren}(\mathcal{C}_{\text{all}}^{(l)}, \boldsymbol{a}_K^{l-1}), \ \mathcal{C}_q^{(l)} = \text{GatherChildren}(\mathcal{C}_{\text{all}}^{(l)}, \boldsymbol{a}_q^{l-1})$$

10:     **2. Similarity computation:**
11:       // compute logits only against the child centroids of the current parent assignment

$$\boldsymbol{\ell}_K^l = \boldsymbol{K}\mathcal{C}_K^{(l)\top}, \quad \boldsymbol{\ell}_q^l = \boldsymbol{q}\mathcal{C}_q^{(l)\top}$$

12:     **3. Local routing probabilities:**
13:       // softmax over the $C$ children of each parent

$$\boldsymbol{p}_K^l = \text{softmax}(\boldsymbol{\ell}_K^l), \quad \boldsymbol{p}_q^l = \text{softmax}(\boldsymbol{\ell}_q^l)$$

14:     **4. Path probability update:**
15:       // update accumulated path probabilities using chosen child probabilities

$$\boldsymbol{P}_K^l = \boldsymbol{P}_K^{l-1} \odot \max_j(\boldsymbol{p}_{K,:,j}^l), \quad \boldsymbol{P}_q^l = \text{BeamUpdateProbs}(\boldsymbol{P}_q^{l-1}, \boldsymbol{p}_q^l, M)$$

16:     **5. Hard assignment (keys: greedy, query: beam):**
17:       // keys: pick the best child index for each token
18:       // query: maintain a beam of $M$ best child indices across all parents

$$a_{K,i}^l = a_{K,i}^{l-1} \cdot C + \arg\max_j \boldsymbol{p}_{K,i,j}^l, \ \boldsymbol{a}_q^l = \text{BeamUpdateIndices}(\boldsymbol{a}_q^{l-1}, \boldsymbol{p}_q^l, M)$$

19: **end for**
20: **Retrieval:** Select top-$M$ buckets based on final query beam probabilities:

$$\mathcal{I} = \text{Top-M-Buckets}(\boldsymbol{P}_q^L, M)$$

21: **Gather:** Construct $\boldsymbol{K}', \boldsymbol{V}'$ by concatenating keys/values from selected buckets $\mathcal{I}$:

$$\boldsymbol{K}' = \text{concat}\{\boldsymbol{K}_i \mid i \in \mathcal{I}\}, \quad \boldsymbol{V}' = \text{concat}\{\boldsymbol{V}_i \mid i \in \mathcal{I}\}$$

22: **Attention:** Compute sparse attention:

$$\boldsymbol{A} = \text{softmax}\left(\boldsymbol{q}\boldsymbol{K}'^\top / \sqrt{d}\right)\boldsymbol{V}'$$

23: **return** $\boldsymbol{A}$

---

**Top-$k$ Attention.** To alleviate the $\mathcal{O}(T^2)$ cost of full self-attention, top-$k$ methods (Kitaev et al., 2020; Roy et al., 2021; Gupta et al., 2021) restrict each query to its $k$ most relevant keys, reducing complexity to $\mathcal{O}(Tk)$. Specifically, for a query vector $\boldsymbol{q} \in \mathbb{R}^d$, let

$$I_q = \text{Top-K}\left(\boldsymbol{q}\boldsymbol{K}^\top / \sqrt{d}, k\right) \tag{2}$$

be the set of indices corresponding to the $k$ largest similarity scores. The attention output is then

$$\text{Attention}_{\text{Top-K}}(\boldsymbol{q}, \boldsymbol{K}, \boldsymbol{V}) = \sum_{i \in I_q} \text{softmax}\left(\boldsymbol{q}\boldsymbol{K}_i^\top / \sqrt{d}\right)\boldsymbol{V}_i. \tag{3}$$

This preserves the expressivity of self-attention while substantially improving efficiency.

## 3.2 HiRouter: Hierarchical Routing for Efficient Top-K Retrieval

We introduce HiRouter, a hardware-efficient method for identifying the top-$k$ relevant key-value pairs. We begin with projected inputs $Z = XW$ (representing either queries $Q$ or keys $K$), where $X \in \mathbb{R}^{B \times T \times d}$, with batch size $B$. The routing process consists of three logical steps: hierarchical definition, discrete routing, and physical layout reorganization.

**Hierarchical Structure.** We organize the routing as a $C$-ary tree with $L$ levels. At any level $l \in \{1, \ldots, L\}$, the search space is partitioned into $C^{l-1}$ parent nodes, where each parent $p$ routes its assigned tokens into one of $C$ child buckets. We define a bank of learnable centroids $\mathcal{C}^{(l)} \in \mathbb{R}^{C^{l-1} \times d \times C}$, where $\mathcal{C}_p^{(l)} \in \mathbb{R}^{d \times C}$ represents $C$ centroids corresponding to the children of parent $p$.

**Routing and Discrete Assignment.** For a token $z$ currently assigned to parent $p$, we compute the routing logits $\ell_p^{(l)} = z\mathcal{C}_p^{(l)}$ and the resulting probability distribution $p_p^{(l)} = \text{softmax}(\ell_p^{(l)})$. To enforce a hard decision while maintaining differentiability, we employ the Gumbel-Softmax trick:

$$\tilde{p}_p^{(l)} = \text{GumbelSoftmax}(\ell_p^{(l)}), \quad a_{\text{local}} = \underset{j \in \{0, \ldots, C-1\}}{\arg\max} \tilde{p}_{p,j}^{(l)} \tag{4}$$

Here, $a_{\text{local}}$ is the local index of the chosen child bucket (relative to parent $p$). The unique global bucket index $a$ for the token at this level is derived by offsetting the parent index: $a = p \cdot C + a_{\text{local}}$.

**Physical Reorganization.** For efficient parallel processing at the next level, tokens must be grouped physically by their assigned buckets. We sort the tokens based on their global indices $\{a_i\}$ such that $\tilde{a}_1 \leq \cdots \leq \tilde{a}_T$. The sorted feature tensor is then reshaped to: $z^{(l+1)} \in \mathbb{R}^{B \times C^l \times \frac{T}{C^l} \times d}$. This reshaping operation treats the $C^l$ buckets as independent batch dimensions for the next hierarchy level $(l+1)$, allowing the routing process to recurse efficiently.

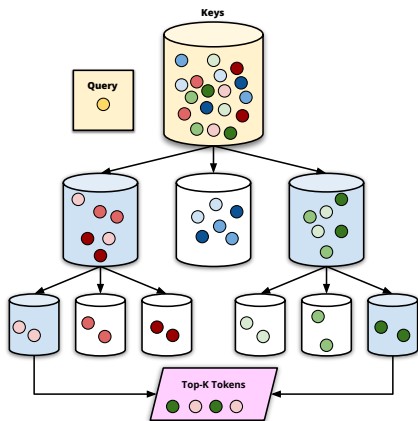

We provide pseudocode in Algorithm 1. The algorithm is fully parallelizable on GPUs: centroid–token similarities at each level are computed for all tokens in parallel, depending only on the feature dimension $d$, giving an $O(d)$ per-token cost. The routing probabilities are then evaluated only over the $C$ children of each parent, which is a constant-size operation that can be parallelized across all parent nodes in $O(C)$. Finally, the hierarchical path probabilities are updated by multiplying each parent's probability with that of its selected child, which requires only $O(L)$ across levels. Overall, every stage of the routing procedure is both parallel and lightweight, enabling HiRouter to be implemented efficiently on modern hardware.

Figure 2: 2-level-HiRouter using beam width 2. Tokens are recursively routed per level. Given a query (yellow dot), the keys and values of previous tokens are aggregated in a buffer. At every level, only the two highest-probability buckets (in blue) at each layer are kept, with the others pruned. Remaining leaves are concatenated into a compact *Top-k* buffer (in pink).

## 3.3 Motivating Entropy Regularization through Router Analysis

To ground our design, we analyze a single router unit and show how its behavior motivates the entropy regularizer that underpins our proposed HiRouter.

**Notation.** Let $z \in \{z_1, \ldots, z_T\} \subset \mathbb{S}^{d-1}$ denote unit–norm tokens and let $\mathcal{C}_j \subset \mathbb{S}^{d-1}$ denote one unit–norm centroid. Each token $z_i$ is assigned to its nearest centroid via $a_i = \arg\max_b \langle z_i, \mathcal{C}_b \rangle$, and satisfies the *intra–bucket tightness*

$$\langle z_i, \mathcal{C}_{a_i} \rangle \geq 1 - \varepsilon, \quad 0 < \varepsilon < 1. \tag{5}$$

Given a query $q \in \mathbb{S}^{d-1}$ with centroid assignment $a_q = \arg\max_b \langle q, \mathcal{C}_b \rangle$, let $\mathcal{S}_q = \{z_i : a_i = a_q\}$ be its bucket. Define the *inter–centroid margin* between $\mathcal{C}_q$ and $\mathcal{C}_{a_z}$ for any $z \notin \mathcal{S}_q$ as $\Delta_{q,z} = 1 - \langle \mathcal{C}_q, \mathcal{C}_{a_z} \rangle$. Obviously, $\Delta_{q,z} \in [0, 2]$.

**Proposition 3.1.** *Let $z^\star = \arg\max_i \langle q, z_i \rangle$ be the nearest neighbor of a query $q \in \mathbb{S}^{d-1}$ among the database $\{z_i\}$. Define $g_{\text{eff}} := \min_{z \notin \mathcal{S}_q}(\langle q, z^\star \rangle - \langle q, z \rangle)$. If $\Delta_{q,z} > \varepsilon + 2\sqrt{2\varepsilon}$ for $z \notin \mathcal{S}_q$. Then $g_{\text{eff}} > 0$ and $z^\star \in \mathcal{S}_q$; i.e., the query $q$ and $z^\star$ are assigned to the same centroid.*

To ensure that a query and its nearest neighbors are assigned to the same bucket by the routers, the centroid margin should exceed the intra–bucket distortion, i.e., $\Delta_{q,z} > \varepsilon + 2\sqrt{2\varepsilon}$. Consequently, learning sharper clusters ($\varepsilon \downarrow$) or achieving more widely separated centroids ($\Delta_{q,z} \uparrow$) directly strengthens the retrieval guarantee for the routers. Our proposed $\mathcal{L}_{\text{smp}}$ encourages key and query embeddings to move toward the bucket centers, thereby shrinking $\varepsilon$ and relaxing the lower bound.

### 3.4 DUAL ENTROPY LOSS

#### 3.4.1 SAMPLE–CENTROID ATTRACTION AND REPULSION

To enforce sharper routing, we apply a Sample-Centroid Loss $\mathcal{L}_{\text{smp}}$, whose gradient naturally decomposes into attractive forces pulling embeddings of keys and queries toward centroids with high assignment probability and repulsive forces pushing them away from low-probability centroids. This attraction–repulsion mechanism progressively aligns embeddings of keys and queries with their most likely centroid while increasing their separation from competing centroids.

Formally, for token $i$ under parent $p$ at the $l$-th level, we define its assignment vector as $p_{i,p}^{(l)} = [p_{i,(p,1)}^{(l)}, \ p_{i,(p,2)}^{(l)}, \ \ldots, \ p_{i,(p,C)}^{(l)}] \in \mathbb{R}^C$, where $p_{i,(p,j)}^{(l)}$ is the probability that the $i$-th token under parent $p$ is routed to its $j$-th child at $l$-th level. The Sample–Centroid Loss is defined as below to sharpen token–centroid alignment:

$$\mathcal{L}_{\text{smp}} = \frac{1}{T} \sum_{i=1}^{T} H(p_{i,p}^{(l)}) = -\frac{1}{T} \sum_{i=1}^{T} \sum_{j=1}^{C} p_{i,(p,j)}^{(l)} \ \log p_{i,(p,j)}^{(l)}. \tag{6}$$

We compute, for each token, the entropy of its assignment distribution at every level of the hierarchical router and average these entropies across tokens and across all levels $l \in \{1, \ldots, L\}$. The resulting loss induces token updates that can be understood through an attraction–repulsion dynamic, as formalized in the following proposition.

**Proposition 3.2.** *At a given router level, let $p$ be the parent node to which token $z_i$ is assigned; denote by $\{\mathcal{C}_{p,j}\}_{j=1}^{C}$ the child centroids under $p$, and by $p_{i,(p,j)}$ the soft assignment probabilities of $z_i$ to those centroids. For the sample-entropy loss $\mathcal{L}_{\text{smp}}^{(p)}$, the gradient w.r.t. $z_i$ is*

$$\nabla_{z_i} \mathcal{L}_{\text{smp}} = -\frac{1}{T} \sum_{j=1}^{C} p_{i,(p,j)} \left( \log p_{i,(p,j)} + 1 \right) \left( \mathcal{C}_{p,j} - \sum_{j'=1}^{C} p_{i,(p,j')} \mathcal{C}_{p,j'} \right).$$

*Hence each centroid $\mathcal{C}_{p,j}$ exerts an attractive effect on $x_i$ iff $p_{i,(p,j)} > e^{-1}$ (since $p(\log p + 1) > 0$), and a repulsive effect iff $p_{i,(p,j)} < e^{-1}$. Thus, the dynamics enforce both intra–bucket tightness ($\varepsilon \downarrow$) and inter–centroid margin ($\Delta_{q,x} \uparrow$), as required by Proposition 3.1.*

As training evolves, the attraction–repulsion dynamics ensure that each embedding of keys and queries is progressively pulled toward its dominant centroid while being pushed away from competing centroids. This dual effect sharpens the assignments, yielding confident one-hot–like routing decisions and enhancing retrieval reliability. Conversely, fractional assignments incur nonzero entropy and therefore generate repulsive forces that enlarge the separation between centroids. Consequently, Proposition 3.2 guarantees the simultaneous decrease of intra–bucket distortion ($\varepsilon \downarrow$) and increase of inter–centroid margin ($\Delta_{q,x} \uparrow$), thereby supporting Proposition 3.1.

#### 3.4.2 BALANCED-ASSIGNMENT LOSS

With only $\mathcal{L}_{\text{smp}}$, keys or queries may collapse into a few centroids, leading to imbalanced bucket sizes. This degrades the parallel efficiency of the underlying computational kernels, as some buckets remain underutilized while others become overloaded. Moreover, with such an imbalance, top-$k$ retrieval becomes inefficient and unstable: some queries retrieve a disproportionately large number of

tokens while others retrieve almost none, resulting in degraded attention performance. To address this, we introduce a balanced-assignment loss that encourages keys or queries to be evenly distributed across centroids, ensuring both statistical robustness and hardware efficiency.

At each parent node $p$ in the hierarchy, every token $i \in \mathcal{I}_p$ must be routed to one of its $C$ children. To ensure balanced routing, we define the mean assignment distribution $\bar{p}_{p,j}^{(l)} = \frac{1}{N_p} \sum_{i \in \mathcal{I}_p} \tilde{p}_{i,(p,j)}^{(l)}, j \in \{0, \ldots, C-1\}$, where $\tilde{p}_{i,(p,j)}^{(l)}$ is the soft assignment of token $i$ to the $j$-th child of parent $p$, and $N_p = |\mathcal{I}_p|$ is the number of tokens under parent $p$. To encourage even splits, we penalize low-entropy mean distributions via the *balanced-assignment loss*:

$$\mathcal{L}_{\mathrm{bal}} = \sum_{p=1}^{C^{l-1}} \sum_{j=1}^{C} \bar{p}_{p,j}^{(l)} \log \bar{p}_{p,j}^{(l)}. \tag{7}$$

Minimizing $\mathcal{L}_{\mathrm{bal}}$ maximizes the entropy of $\bar{\boldsymbol{p}}_p$, driving each $\bar{p}_{p,j}$ toward the uniform distribution $[1/C, \ldots, 1/C]$. This ensures that tokens are spread evenly across the $C$ children, enabling contiguous tensor reshaping and efficient parallel operations.

**Proposition 3.3.** *Under parent $p$, let $\{\tilde{\boldsymbol{p}}_{i,(p,j)}\}_{i \in \mathcal{I}_p}$ and define $\bar{\boldsymbol{p}}_{p,j} = \frac{1}{N_p} \sum_{i \in \mathcal{I}_p} \tilde{\boldsymbol{p}}_{i,(p,j)}$, with $\sum_{j=1}^{C} \bar{p}_{p,j} = 1$. The balanced loss is minimized iff $\bar{p}_{p,j} = 1/C$ for all $j$.*

**Why Gumbel–Softmax (GS)?** Using a vanilla $\mathrm{softmax}$ to parameterize assignments makes the gradient of $\mathcal{L}_{\mathrm{bal}}$ identical for all tokens under the same parent, pushing every token's distribution toward the uniform vector $[1/C]^C$. This maximizes per-token entropy, contradicting the sample-entropy loss $\mathcal{L}_{\mathrm{smp}}$, leading to ambiguous assignments and poorly balanced buckets. GS, with a straight-through estimator (Jang et al., 2017), instead produces near one-hot assignments while remaining differentiable. This allows each token to select a single centroid, so that minimizing $\mathcal{L}_{\mathrm{bal}}$ balances the counts $N_{p,j}$ across children, ensuring roughly uniform bucket populations and enabling efficient top-$k$ attention kernels without sacrificing gradient flow.

### 3.4.3 OVERALL ROUTING OBJECTIVE.

The final routing loss combines both terms: $\mathcal{L}_{\mathrm{route}} = \mathcal{L}_{\mathrm{bal}} + \mathcal{L}_{\mathrm{smp}}$. In practice, $\mathcal{L}_{\mathrm{route}}$ serves to regularize the query/key projections, promoting an emergent hierarchical representation space. We integrate the routing regularizer with the downstream task loss, such that the total objective becomes $\mathcal{L} = \mathcal{L}_{\mathrm{CE}} + \alpha \mathcal{L}_{\mathrm{route}}$, where $\mathcal{L}_{\mathrm{CE}} = -\frac{1}{T} \sum_{t=1}^{T} \log P(y_t \mid y_{<t})$ is the standard next-token cross-entropy, $\mathcal{L}_{\mathrm{route}}$ is our hierarchical routing loss, and $\alpha > 0$ controls the regularization strength.

### 3.5 CANDIDATE RETRIEVAL FOR ATTENTION

The router assigns each token to a bucket at each hierarchical level according to its routing probability. We first project them via learned matrices $\boldsymbol{W}_Q, \boldsymbol{W}_K$ and compute their similarity:

$$\boldsymbol{S} = \left(\boldsymbol{X}\boldsymbol{W}_Q^\top\right) \left(\boldsymbol{X}\boldsymbol{W}_K^\top\right)^\top.$$

At level $l$, each token has a conditional routing distribution $p^{(l)} \in \mathbb{R}^C$.

A brute-force strategy would enumerate all buckets with nonzero joint probability $p_{\mathrm{joint}} = \prod_{l=0}^{L-1} p^{(l)}$, but this quickly becomes prohibitive in both time and memory as there exists a total $\prod_{l=0}^{L-1} C^l$ possibilities. Instead, we perform a level-wise beam search of width $M$: at each level $l$, we retain the $M$ buckets with largest partial joint probability $\prod_{i=0}^{l} p^{(i)}$. These $M$-element beams define a candidate set of key tokens for retrieval. To compute attention outputs, we collect these $M$ buckets and then compute sparse attention following Equation 3.

## 4 EXPERIMENTS

### 4.1 SYNTHETIC RESULTS

**Synthetic Gaussian Retrieval.** To validate the efficiency and recall of HIROUTER, we first evaluate on a synthetic key–value retrieval task. We generate $N$ keys and values by sampling from a standard

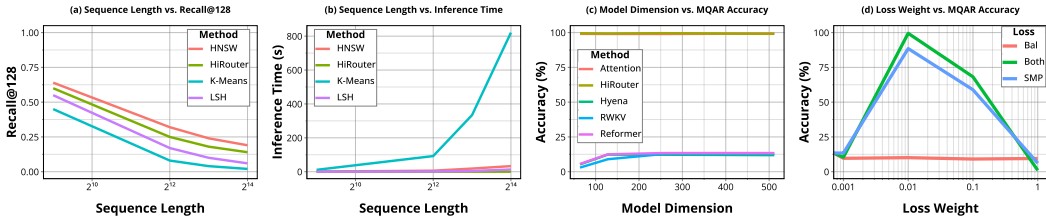

Figure 3: Synthetic experimental results, demonstrating the efficacy of HιRоUTER. (a) and (b) show performance relative to contemporary retrieval methods in terms of recall and speed with relation to the sequence length $T$. (c) plot shows the effect of hidden size $d$ on MQAR's accuracy, while (d) plot shows how the scale $\alpha$ of the auxiliary loss can influence performance on MQAR.

multivariate Gaussian in $\mathbb{R}^d$, insert them into our hierarchical router, and then issue $M$ random key queries drawn from the same distribution, similar setting in (Kraska et al., 2018). We measure recall@128 (i.e. the fraction of queries whose top-128 retrieved key matches the true maximum inner production keys) and end-to-end latency as we sweep $N$ from $2^{12}$ to $2^{18}$. As shown in Figure 3(b), HιRоUTER maintains recall even higher than LSH and vanilla $k$-means. HNSW achieves slightly higher recall, but its query time increases superlinearly, resulting in prohibitive latency for processing long sequences in Attention. Baselines are implemented using FAISS (Douze et al., 2024).

**Multi-Query Associative Recall (MQAR).** Next, we benchmark on the MQAR task (Arora et al., 2024) where the model must store a sequence of $N$ key–value pairs and then retrieve the correct value given a set of query keys. The total vocab size is 8192. Figure 3(c) shows that HιRоUTER sustains high recall even for small $d$, whereas other methods fail. Finally, we sweep the weight $\alpha$ on our dual-entropy routing loss. As shown in Figure 3(d), choosing $\alpha$ in $[0.01, 0.1]$ yields the best trade-off: too small an $\alpha$ leaves $\mathcal{L}_{\text{bal}}$ ineffective, while a large $\alpha$ (in the absence of $\mathcal{L}_{\text{smp}}$) allows trivial uniform assignments that destroys semantic clustering and hurts recall.

## 4.2 SMALL SCALE LANGUAGE MODELING

We first compare the performance of a HιRоUTER enhanced Transformer on a classic language modeling task, namely WikiText-103 language modeling. In this setting, we use $\alpha$ as we determined best on the MQAR task. All models used in this task are configured with 125M parameters. Our primary observation is that HιRоUTER outperforms the standard Transformer, achieving a 0.7 reduction in perplexity; we achieve better perplexity alongside a significant efficiency improvement. Additionally, alternative efficient attention methods observe a significant degradation, highlighting that HιRоUTER can serve as a better choice for efficient Transformers.

Table 1: Test perplexity (lower is better) on WikiText-103.

| Model | ppl ↓ |
|---|---|
| Transformer | 19.2 |
| Performer | 26.8 |
| Reformer | 25.6 |
| AFT-conv | 28.2 |
| RFA-Gaussian | 27.5 |
| CosFormer | 23.1 |
| IceFormer | 31.4 |
| Routing Tranformer | 26.7 |
| BigBird | 23.3 |
| ZETA | 26.3 |
| Mongoose | 23.6 |
| NSA | 19.3 |
| HιRоUTER | **18.5** |

## 4.3 LARGER SCALE LANGUAGE MODELING

**Setup and Training.** We conduct an evaluation of our method against other methods, such as a Transformer based on the `Pythia` architecture (Biderman et al., 2023)[1] as well as RetNet (Sun et al., 2023), Mamba (Gu & Dao, 2024; Dao & Gu, 2024), Gated Linear Attention (GLA) (Yang et al., 2024a), DeltaNet (Yang et al., 2024b), Gated Slot Attention (GSA) (Zhang et al., 2024). For fair comparison, all models are trained under identical conditions with 410M parameters on 10B tokens from the `FineWeb-Edu` dataset (Penedo et al., 2024), with some restrictions[2]. All models are trained with a context length of 2048 tokens, with embedding/hidden dimension 1024. We use the AdamW optimizer (Loshchilov & Hutter, 2019) with a peak learning rate of 4e-4, weight decay of 0.1, and gradient clipping of 1.0. The learning rate follows a cosine annealing schedule with a warm-up period of 1% of the total steps ($\approx$100M tokens) and a total batch size of 0.5M tokens. Further details are available in Appendix B. For our HιRоUTER model, we use the same training setup and configure our router as having 4 levels, each level with 4 buckets/centroids, the

---

[1]Some works follow Gu & Dao (2024) and refer to this architecture as `Transformer++`.

[2]Mamba models use $\approx$430M parameters due to restrictions on the state size and the input dimension.

window size as $64$ for the SWA branch, and the top-$k$ attention using a beam width of 4. Following our results on the synthetic task, we choose $\alpha$ to be 0.05 to set the weight for the auxiliary loss.

### 4.3.1 COMMONSENSE REASONING

Table 2: Performance comparison on language modeling and zero-shot common-sense reasoning.

| Model | Wiki. ppl ↓ | LMB. ppl ↓ | LMB. acc ↑ | PIQA acc ↑ | Hella. acc_n ↑ | Wino. acc ↑ | ARC-e acc ↑ | ARC-c acc_n ↑ | SIQA acc ↑ | BoolQ acc ↑ | Avg. |
|---|---|---|---|---|---|---|---|---|---|---|---|
| Transformer | 30.21 | 43.44 | 32.76 | **67.68** | 39.20 | **53.51** | 57.58 | 27.47 | 37.97 | 61.19 | **47.17** |
| RetNet | 36.47 | 63.64 | 26.33 | 65.18 | 35.61 | 50.59 | 56.82 | 27.13 | 37.87 | 60.95 | 45.06 |
| Mamba | 32.63 | 61.68 | 27.79 | 65.61 | 38.47 | 51.22 | 57.41 | 26.62 | 38.64 | **61.65** | 45.93 |
| Mamba2 | 30.15 | 49.83 | 28.70 | 66.81 | 38.94 | 51.38 | **60.69** | 28.75 | 37.67 | 59.69 | 46.83 |
| HGRN2 | 30.07 | **40.29** | 31.32 | 66.54 | **39.68** | 50.12 | 59.30 | 27.05 | 38.84 | 58.72 | 46.45 |
| GLA | 31.50 | 51.56 | 29.01 | 66.49 | 38.60 | 50.12 | 57.83 | 26.11 | **39.25** | 57.77 | 45.40 |
| DeltaNet | **28.82** | 45.06 | 30.47 | 67.19 | 39.51 | 52.80 | 58.80 | **29.10** | 38.18 | 58.26 | 46.79 |
| GSA | 30.78 | 48.74 | 29.75 | 66.70 | 39.01 | 52.49 | 59.26 | 27.65 | 38.49 | 60.61 | 46.50 |
| HeadKV | 34.01 | 43.67 | 27.29 | 67.43 | 37.86 | 51.62 | 58.34 | 24.23 | 37.82 | 61.77 | 45.80 |
| SqueezeAttn | 34.00 | 43.27 | 30.78 | 66.21 | 38.99 | 51.30 | 59.30 | 27.56 | 37.46 | 61.53 | 46.64 |
| ClusterKV | 33.01 | 45.67 | 30.51 | 65.41 | 36.75 | 49.82 | 58.92 | 26.51 | 37.17 | 60.12 | 45.65 |
| HIROUTER | 31.09 | 42.94 | **33.09** | 66.81 | 38.03 | 50.75 | 59.47 | 28.50 | 38.08 | 61.31 | 47.01 |

Similar to previous works, we present perplexity results as well as zero-shot commonsense reasoning performance on a number of different tasks (see Appendix B.4.1). These tasks are effective at evaluating the acquired knowledge of models through their general reasoning abilities. In Table 2, we observe that HIROUTER is effective in comparison to a number of modern methods commonly used as efficient language model backbones. In particular, we observe that while a baseline, full-attention Transformer remains the most effective model compared to other alternatives, HIROUTER remains highly effective on such tasks and performs comparably or outperforms recent models that offer efficiency gains in comparison to the Transformer.

### 4.3.2 RECALL-INTENSIVE TASKS

To better compare the ability of models to recall information, we evaluate zero-shot in-context learning performance on more recall-intensive tasks (Appendix B.4.2). As shown in Table 3, the Transformer fares best, while other efficient baselines generally struggle due to their fixed-size state. In contrast, HIROUTER remains capable of on-par performance relative to the Transformer while maintaining efficiency. This re-

Table 3: Accuracy on recall-world retrieval tasks.

| Model | FDA | SWDE | SQuAD | TQA | NQ | Drop | Avg. |
|---|---|---|---|---|---|---|---|
| Transformer | 7.26 | 38.07 | 4.52 | 0.93 | 1.00 | **2.48** | 9.71 |
| RetNet | 0.02 | 0.02 | **46.12** | 0.02 | 0.06 | 0.02 | 7.70 |
| Mamba | 1.36 | 6.84 | 3.10 | **1.03** | 1.00 | 2.12 | 2.91 |
| Mamba2 | 4.26 | 10.53 | 4.49 | 0.55 | 1.25 | 2.43 | 3.92 |
| HGRN2 | 2.00 | 10.17 | 4.02 | 0.97 | 1.02 | 3.12 | 3.88 |
| GLA | 3.27 | 9.72 | 2.72 | 0.40 | 1.36 | 1.96 | 3.24 |
| DeltaNet | 4.08 | 17.19 | 3.81 | 0.42 | 0.97 | 2.41 | 4.81 |
| GSA | 3.36 | 7.02 | 4.13 | 0.86 | **1.47** | 2.47 | 3.55 |
| HIROUTER | **8.43** | **42.83** | 3.38 | 0.67 | 0.93 | 2.32 | **9.76** |

sults demonstrates a use-case where the HIROUTER structure can potentially serve as beneficial for filtering out irrelevant information.

### 4.3.3 LONG-CONTEXT TASKS

Finally, we test on LongBench (Bai et al., 2024), a common benchmark for evaluating performance on long-context tasks (see Appendix B.4.3). In this setting, shown in Table 4, Transformers struggle, reflecting a long-standing observation regarding the inability of full-attention models to adequately manipulate long sequences. Meanwhile, linear models are much more performant. In comparison, we show that HIROUTER is capable of significantly closing the gap between these two paradigms, highlighting the potential for improved long-context Transformer models, being able to outperform other baselines outside of Mamba even without additional tuning of the model parameters.

Additionally, we perform a synthetic evaluation on the Needle-in-a-Haystack (NIAH) task, where models are tasked with retrieving a single element (the needle) from a large context (the haystack). Table 5 presents these results. It is worth noting that Transformers are generally much more effective on context lengths within the scope of the training context, highlighted by strong performance in different formats of the needle within the haystack. However, some recurrent models demonstrate a

Table 4: Accuracy on tasks from LongBench (Bai et al., 2024).

| Model | Single-Doc QA | | | Multi-Doc QA | | | Summarization | | | Few-shot | | | Code | | Avg. |
|---|---|---|---|---|---|---|---|---|---|---|---|---|---|---|---|
| | NQA | QQA | MFQ | HQA | 2WM | Mus | GvR | QMS | MNs | TRC | TQA | SSM | LCC | RBP | |
| Transformer | 0.67 | 3.23 | 3.86 | 0.33 | 1.37 | 0.11 | 8.29 | 11.87 | **13.31** | 1.50 | 3.02 | 5.61 | 9.82 | 9.61 | 5.19 |
| HGRN2 | 0.38 | 0.80 | 1.63 | 0.11 | 0.05 | 0.11 | 3.07 | 5.96 | 4.08 | 0.00 | 0.67 | 0.00 | **20.88** | **20.71** | 4.17 |
| Mamba | 1.52 | **3.55** | 10.51 | 3.20 | **6.82** | 2.24 | 5.51 | 15.67 | 10.02 | 3.00 | **14.04** | 5.82 | 11.55 | 14.82 | 7.73 |
| Mamba2 | **1.80** | 3.20 | 10.84 | 2.97 | 5.70 | 2.57 | 6.66 | 15.87 | 10.43 | 18.50 | 13.31 | 6.09 | 16.67 | 19.05 | **9.55** |
| GLA | 0.60 | 1.46 | 2.50 | 0.72 | 1.01 | 0.70 | 4.30 | 10.44 | 6.41 | 0.00 | 5.58 | 0.00 | 20.23 | 20.45 | 5.31 |
| DeltaNet | 0.38 | 0.76 | 1.63 | 0.11 | 0.05 | 0.11 | 3.21 | 7.40 | 4.50 | 0.00 | 5.58 | **9.30** | 20.03 | 19.89 | 5.21 |
| GSA | 0.37 | 0.73 | 1.60 | 0.11 | 0.05 | **4.81** | 3.28 | 8.61 | 4.81 | 0.00 | 4.71 | 8.27 | 19.33 | 20.16 | 5.49 |
| HIROUTER | 1.69 | 3.54 | **11.25** | **4.54** | 6.82 | 2.54 | **8.80** | **16.21** | 10.66 | **21.17** | 11.36 | 4.48 | 5.25 | 11.21 | 8.54 |

Table 5: Zero-shot performance on S-NIAH tasks from RULER (Hsieh et al., 2024).

| Model | S-NIAH-1 (pass-key retrieval) | | | | S-NIAH-2 (number in haystack) | | | | S-NIAH-3 (uuid in haystack) | | | | Avg. |
|---|---|---|---|---|---|---|---|---|---|---|---|---|---|
| | 1K | 2K | 4K | 8K | 1K | 2K | 4K | 8K | 1K | 2K | 4K | 8K | |
| Transformer | **94.8** | **96.0** | 0.0 | 0.0 | **95.6** | **70.8** | 0.0 | 0.0 | **91.6** | 57.6 | 0.0 | 0.0 | 42.2 |
| GLA | 0.0 | 0.0 | 0.0 | 0.0 | 3.2 | 1.6 | 1.2 | 0.8 | 0.0 | 0.0 | 0.0 | 0.0 | 0.6 |
| HGRN2 | 76.0 | 4.8 | 0.0 | 0.0 | 36.4 | 7.6 | 0.0 | 0.0 | 0.0 | 0.0 | 0.0 | 0.0 | 10.4 |
| Mamba | 8.8 | 4.0 | 1.2 | 0.8 | 27.2 | 3.6 | 2.4 | 2.4 | 0.0 | 0.0 | 0.0 | 0.0 | 4.2 |
| Mamba2 | 35.2 | 9.6 | 0.8 | 0.0 | 25.2 | 6.4 | 11.6 | 1.6 | 0.8 | 1.6 | 0.4 | 0.0 | 7.7 |
| DeltaNet | 38.8 | 40.8 | 48.4 | **34.8** | 26.4 | 6.0 | 10.8 | 4.4 | 8.0 | 0.8 | 0.8 | **2.4** | 18.5 |
| GSA | 23.6 | 10.0 | 3.2 | 2.4 | 20.4 | 6.8 | 9.2 | **4.8** | 0.0 | 0.0 | 0.0 | 0.0 | 6.7 |
| HIROUTER | 93.6 | 86.8 | **57.4** | 22.4 | 84.2 | 67.6 | **32.6** | 4.4 | 88.4 | **60.4** | **22.2** | 2.4 | 51.9 |

better propensity to extrapolate beyond the training context, such as Mamba, DeltaNet, and GSA. HIROUTER again demonstrates the ability to bridge this gap in effectiveness between these two paradigms: on shorter contexts, the performance remains comparable to the initial Transformer, but as the context length extends, HIROUTER retains an ability to extrapolate and still perform at par with models specifically trained for long contexts and extrapolation.

## 4.4 COMPUTATIONAL EFFICIENCY EXPERIMENT

To further quantify HIROUTER's runtime behavior, we benchmark it alongside two sparse top-$k$ attention baselines, Routing Transformer and Mongoose, on a fixed batch size while scaling sequence length $T$. Table 6 reports forward (FWD) and forward+backward (FWD+BWD) runtimes (ms) for sequence lengths between $2^{12}$ and $2^{16}$. HIROUTER consistently outpaces both Routing Transformer and Mongoose in forward/backward modes while surpassing their scaling behavior: HIROUTER remains efficient at the largest tested lengths while others do not.

Table 6: Time (in milliseconds) for forward (FWD) and forward+backward (FWD+BWD) passes on a fixed-sized batch across varying sequence lengths. Lower is better.

| Input Length | FlashAttention | | Routing Transformer | | Mongoose | | HIROUTER | |
|---|---|---|---|---|---|---|---|---|
| | FWD | FWD+BWD | FWD | FWD+BWD | FWD | FWD+BWD | FWD | FWD+BWD |
| 4096 | **0.18** | **0.61** | 2.88 | 5.81 | 2.21 | 4.58 | 0.26 | 0.75 |
| 8192 | **0.56** | 1.95 | 3.76 | 8.20 | 3.76 | 8.20 | 0.59 | **1.81** |
| 16384 | 1.93 | 6.55 | 7.98 | 18.58 | 6.76 | 15.04 | **1.68** | **6.08** |
| 32768 | 7.14 | 25.09 | 15.76 | 36.08 | 13.29 | 29.08 | **3.12** | **16.81** |
| 65536 | 30.76 | 99.69 | 33.79 | 73.41 | 29.66 | 60.34 | **8.11** | **40.01** |

## 5 CONCLUSION

In this work, we present HIROUTER, a novel hierarchical routing approach towards computing top-$k$ attention via maximum inner product search. HIROUTER uses a bucket partitioning approach, partitioning tokens within the sequence into discrete buckets across multiple levels of a learned tree. The tree uses learned centroid-based routing logits and a Gumbel-Softmax trick with a dual-component routing loss for training. Our work provides empirical evidence to show that HIROUTER is both competitive with concurrent efficient LLM architectures as well as regular full-attention baselines. Furthermore, we provide an efficient Trition-based implementation to enable our method to outperform other efficient attention-based implementations in terms of efficiency.

## ETHICS STATEMENT

This work focuses on developing efficient attention mechanisms for large-scale language models. While our method improves the scalability and retrieval accuracy of Transformer models, the ethical considerations are largely consistent with those of general-purpose language modeling. Potential risks include misuse in generating harmful or misleading content, reinforcement of biases present in training corpora, and environmental concerns arising from large-scale training. We mitigate these risks by (i) benchmarking only on standard public datasets, (ii) avoiding the use of sensitive or private data, and (iii) providing transparent methodology to facilitate responsible replication. Moreover, the computational efficiency gains of HIROUTERreduce energy consumption relative to dense or less efficient sparse baselines, contributing positively to the environmental impact of model deployment.

## REPRODUCIBILITY STATEMENT

We have taken steps to ensure reproducibility and transparency in all aspects of this work. The proposed HIROUTER algorithm, including hierarchical routing, dual entropy regularization, and beam-search retrieval, is fully described in the methodology section with precise mathematical formulations. Detailed hyperparameter choices, model architectures, and training procedures are provided in the appendix, including dataset splits, optimization settings, and auxiliary loss scaling. Synthetic experiments, WikiText-103 evaluations, and large-scale benchmarks are reported with sufficient detail to enable replication. We also release a Triton-based implementation of our GPU kernels, ensuring that researchers can reproduce both the efficiency and accuracy results.

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

# Appendix

## A  THEORETICAL ANALYSIS AND PROOFS

**Notation.**  Let $\{z_1, \ldots, z_T\} \subset \mathbb{S}^{d-1}$ denote unit–norm tokens and let $\{\mathcal{C}_1, \ldots, \mathcal{C}_C\} \subset \mathbb{S}^{d-1}$ be unit–norm centroids. Each token $z_i$ is assigned to its nearest centroid

$$a_i = \arg\max_b \langle z_i, \mathcal{C}_b \rangle,$$

and satisfies the *intra–bucket tightness*

$$\langle z_i, \mathcal{C}_{a_i} \rangle \geq 1 - \varepsilon, \quad 0 < \varepsilon < 1. \tag{1}$$

Given a query $q \in \mathbb{S}^{d-1}$ with centroid assignment $a_q = \arg\max_b \langle q, \mathcal{C}_b \rangle$, let $\mathcal{S}_q = \{z_i : a_i = a_q\}$ be its bucket. Define the *inter–centroid margin* between $\mathcal{C}_q$ and $\mathcal{C}_{a_z}$ for any $z \notin \mathcal{S}_q$ as

$$\Delta_{q,z} = 1 - \langle \mathcal{C}_q, \mathcal{C}_{a_z} \rangle, \quad \Delta_{q,z} \in [0, 2]. \tag{2}$$

**Lemma A.1** (Tight cluster). *For any $z_i, z_j \in \mathcal{S}_q$,*

$$\langle z_i, z_j \rangle \geq 1 - 4\varepsilon.$$

*Proof.* By Equation (1), $\|z - \mathcal{C}_q\| \leq \sqrt{2\varepsilon}$ for each $z \in \mathcal{S}_q$. Thus $\|z_i - z_j\| \leq 2\sqrt{2\varepsilon}$, and since both are unit–norm, $\langle z_i, z_j \rangle = 1 - \frac{1}{2}\|z_i - z_j\|^2 \geq 1 - 4\varepsilon$. $\square$

**Lemma A.2** (Residual norm bound). *If $\|q\| = \|\mathcal{C}_q\| = 1$ and $\langle q, \mathcal{C}_q \rangle \geq 1 - \varepsilon$, then in the decomposition $q = \langle q, \mathcal{C}_q \rangle \mathcal{C}_q + r$, with $r \perp \mathcal{C}_q$, we have*

$$\|r\| \leq \sqrt{2\varepsilon}.$$

*Proof.* $\|q\|^2 = \langle q, \mathcal{C}_q \rangle^2 + \|r\|^2$, so $\|r\|^2 = 1 - \langle q, \mathcal{C}_q \rangle^2 = (1 - \langle q, \mathcal{C}_q \rangle)(1 + \langle q, \mathcal{C}_q \rangle)$. Since $\langle q, \mathcal{C}_q \rangle \geq 1 - \varepsilon$, it follows that $\|r\|^2 \leq 2\varepsilon$. Taking square roots yields the claimed bound. $\square$

**Lemma A.3** (Orthogonal component bound). *If $\|\mathcal{C}_q\| = \|\mathcal{C}_{a_z}\| = 1$ and $\langle \mathcal{C}_q, \mathcal{C}_{a_z} \rangle = 1 - \Delta_{q,z}$, then for $\mathcal{C}_{a_z}^\perp = \mathcal{C}_{a_z} - \langle \mathcal{C}_q, \mathcal{C}_{a_z} \rangle \mathcal{C}_q$,*

$$\|\mathcal{C}_{a_z}^\perp\| \leq \sqrt{2\Delta_{q,z}}.$$

*Proof.* Because $\mathcal{C}_{a_z}^\perp \perp \mathcal{C}_q$ and $\|\mathcal{C}_{a_z}\| = \|\mathcal{C}_q\| = 1$,

$$\|\mathcal{C}_{a_z}^\perp\|^2 = 1 - \langle \mathcal{C}_q, \mathcal{C}_{a_z} \rangle^2 = 1 - (1 - \Delta_{q,z})^2 = 2\Delta_{q,z} - \Delta_{q,z}^2 \leq 2\Delta_{q,z}.$$

Taking square roots yields the desired inequality. $\square$

**Lemma A.4** (Centroid gap with distortion). *If $\langle q, \mathcal{C}_q \rangle \geq 1 - \varepsilon$ and $\langle \mathcal{C}_q, \mathcal{C}_{a_z} \rangle = 1 - \Delta_{q,z}$ with $\Delta_{q,z} > \varepsilon$, then*

$$\langle q, \mathcal{C}_{a_z} \rangle \leq \langle q, \mathcal{C}_q \rangle - (\Delta_{q,z} - \varepsilon).$$

*Proof.* Decompose $q = \langle q, \mathcal{C}_q \rangle \mathcal{C}_q + r$ with $\|r\| \leq \sqrt{2\varepsilon}$ (Lemma A.2), and let $\mathcal{C}_{a_z}^\perp$ be from Lemma A.3. Then

$$\langle q, \mathcal{C}_{a_z} \rangle = \langle q, \mathcal{C}_q \rangle (1 - \Delta_{q,z}) + \langle r, \mathcal{C}_{a_z}^\perp \rangle \leq \langle q, \mathcal{C}_q \rangle (1 - \Delta_{q,z}) + \sqrt{2\varepsilon}\sqrt{2\Delta_{q,z}}.$$

Since $\sqrt{2\varepsilon}\sqrt{2\Delta_{q,z}} \leq \Delta_{q,z} - \varepsilon$, the result follows. $\square$

**Proposition A.5.** *Let*

$$z^\star = \arg\max_i \langle q, z_i \rangle$$

*be the true nearest neighbor of query $q \in \mathbb{S}^{d-1}$ among $\{z_i\}$, and let $\mathcal{S}_q$ be the bucket of $q$. Define the effective gap*

$$g_{\mathrm{eff}} = \min_{z \notin \mathcal{S}_q} \left( \langle q, z^\star \rangle - \langle q, z \rangle \right).$$

*If $\Delta_{q,z} > \varepsilon + 2\sqrt{2\varepsilon}$ for all $z \notin \mathcal{S}_q$, then $g_{\mathrm{eff}} > 0$ and $z^\star \in \mathcal{S}_q$; i.e., the query and its nearest neighbor are assigned to the same centroid.*

*Proof.* **1. Bound on $\|z - \mathcal{C}_{a_z}\|$.** For any $z \notin \mathcal{S}_q$, its assigned centroid $\mathcal{C}_{a_z}$ satisfies $\langle z, \mathcal{C}_{a_z} \rangle \geq 1 - \varepsilon$. Since $\|z\| = \|\mathcal{C}_{a_z}\| = 1$,

$$\|z - \mathcal{C}_{a_z}\|^2 = 2\left(1 - \langle z, \mathcal{C}_{a_z} \rangle\right) \leq 2\varepsilon \implies \|z - \mathcal{C}_{a_z}\| \leq \sqrt{2\varepsilon}.$$

**2. Outsider score upper bound.** Because $\|q\| = 1$, Cauchy–Schwarz gives $|\langle q, z - \mathcal{C}_{a_z} \rangle| \leq \sqrt{2\varepsilon}$. Thus

$$\langle q, z \rangle = \langle q, \mathcal{C}_{a_z} \rangle + \langle q, z - \mathcal{C}_{a_z} \rangle \leq \langle q, \mathcal{C}_{a_z} \rangle + \sqrt{2\varepsilon}.$$

By Lemma A.4, $\langle q, \mathcal{C}_{a_z} \rangle \leq \langle q, \mathcal{C}_q \rangle - (\Delta_{q,z} - \varepsilon)$, so

$$\boxed{\langle q, z \rangle \leq \langle q, \mathcal{C}_q \rangle - (\Delta_{q,z} - \varepsilon) + \sqrt{2\varepsilon} = \langle q, \mathcal{C}_q \rangle - \left(\Delta_{q,z} - \varepsilon - \sqrt{2\varepsilon}\right).} \tag{A}$$

**3. Insider score lower bound.** By intra–bucket tightness,

$$\|z^\star - \mathcal{C}_q\| \leq \sqrt{2\varepsilon}.$$

Since $\|q\| = 1$, Cauchy–Schwarz gives

$$|\langle q, z^\star - \mathcal{C}_q \rangle| \leq \|z^\star - \mathcal{C}_q\| \leq \sqrt{2\varepsilon}.$$

Therefore,

$$\langle q, z^\star \rangle = \langle q, \mathcal{C}_q \rangle + \langle q, z^\star - \mathcal{C}_q \rangle \geq \langle q, \mathcal{C}_q \rangle - \sqrt{2\varepsilon}. \tag{B}$$

**4. Effective gap.** Subtracting (A) from (B) yields, for every $z \notin \mathcal{S}_q$,

$$\langle q, z^\star \rangle - \langle q, z \rangle \geq (\Delta_{q,z} - \varepsilon) - 2\sqrt{2\varepsilon} = \Delta_{q,z} - \left(\varepsilon + 2\sqrt{2\varepsilon}\right).$$

Hence

$$g_{\text{eff}} = \min_{z \notin \mathcal{S}_q}\{\langle q, z^\star \rangle - \langle q, z \rangle\} \geq \Delta_{q,z} - \left(\varepsilon + 2\sqrt{2\varepsilon}\right).$$

**5. Correct assignment of $z^\star$ and $q$.** If $\Delta_{q,z} > \varepsilon + 2\sqrt{2\varepsilon}$ then $g_{\text{eff}} > 0$, so $z^\star$ scores strictly above every outsider. As it is also the top insider, it must lie in the same bucket as $q$. $\qquad\square$

**Proposition A.6.** *Under parent $p$, let $\{\tilde{p}_{i,(p,j)}\}_{i \in \mathcal{I}_p}$ and define $\bar{p}_{p,j} = \frac{1}{N_p}\sum_{i \in \mathcal{I}_p}\tilde{p}_{i,(p,j)}$, with $\sum_{j=1}^C \bar{p}_{p,j} = 1$. The balanced loss is minimized iff $\bar{p}_{p,j} = 1/C$ for all $j$.*

*Proof.* Fix parent $p$ and write $\bar{p}_j := \bar{p}_{p,j}$. Introduce a Lagrange multiplier $\lambda$ for the constraint $\sum_{j=1}^C \bar{p}_j = 1$:

$$\mathcal{L}(\{\bar{p}_j\}, \lambda) = \sum_{j=1}^C \bar{p}_j \log \bar{p}_j + \lambda\left(\sum_{j=1}^C \bar{p}_j - 1\right).$$

Stationarity $\partial \mathcal{L}/\partial \bar{p}_j = 0$ gives $\log \bar{p}_j + 1 + \lambda = 0$, so $\bar{p}_j = e^{-(\lambda+1)}$. Enforcing $\sum_{j=1}^C \bar{p}_j = Ce^{-(\lambda+1)} = 1$ yields $\bar{p}_j = 1/C$ for all $j$, the unique minimizer of $\mathcal{L}_{\text{bal}}$.

Under a low-temperature Gumbel–Softmax, each $\tilde{p}_{i,(p,\cdot)}$ is nearly one-hot, so $\bar{p}_{p,j}$ converges to the fraction of tokens assigned to bucket $j$. Driving $\bar{p}_p \to (1/C, \ldots, 1/C)$ thus enforces an approximately equal token count per bucket. $\qquad\square$

**Proposition A.7.** *At a given router level, let $p$ be the parent node to which token $z_i$ is assigned; denote by $\{\mathcal{C}_{p,j}\}_{j=1}^C$ the child centroids under $p$, and by $p_{i,(p,j)}$ the soft assignment probabilities of $z_i$ to those centroids. For the sample-entropy loss $\mathcal{L}_{\text{smp}}^{(p)}$, the gradient w.r.t. $z_i$ is*

$$\nabla_{z_i} \mathcal{L}_{\text{smp}} = -\frac{1}{T}\sum_{j=1}^C p_{i,(p,j)}\left(\log p_{i,(p,j)} + 1\right)\left(\mathcal{C}_{p,j} - \sum_{j'=1}^C p_{i,(p,j')}\mathcal{C}_{p,j'}\right).$$

*Hence each centroid $\mathcal{C}_{p,j}$ exerts an* attractive *effect on $x_i$ iff $p_{i,(p,j)} > e^{-1}$ (since $p(\log p + 1) > 0$), and a* repulsive *effect iff $p_{i,(p,j)} < e^{-1}$. Thus, the dynamics enforce both intra–bucket tightness ($\varepsilon \downarrow$) and inter–centroid margin ($\Delta_{q,x} \uparrow$), as required by Proposition 3.1.*

*Proof.* For a fixed token index $i$, abbreviate

$$p_j := p_{i,(p,j)}, \qquad \boldsymbol{C}_j := \mathbf{C}_{p,j}, \qquad \ell_j := \mathbf{z}_i^\top \boldsymbol{C}_j.$$

Then $p_j = \mathrm{softmax}(\ell)_j = \exp(\ell_j)/\sum_{k=1}^C \exp(\ell_k)$ and the per-sample entropy term is

$$\mathcal{L}_i = -\sum_{j=1}^C p_j \log p_j, \qquad \mathcal{L}_{\mathrm{smp}}^{(p)} = \frac{1}{T} \sum_{i=1}^T \mathcal{L}_i.$$

Differentiating $\mathcal{L}_i$ with respect to $\mathbf{z}_i$ and using $\nabla_{\mathbf{z}_i}(p_j \log p_j) = (\log p_j + 1) \nabla_{\mathbf{z}_i} p_j$ gives

$$\nabla_{\mathbf{z}_i} \mathcal{L}_{\mathrm{smp}}^{(p)} = \frac{1}{T} \nabla_{\mathbf{z}_i} \mathcal{L}_i = -\frac{1}{T} \sum_{j=1}^C (\log p_j + 1) \nabla_{\mathbf{z}_i} p_j.$$

By the softmax Jacobian,

$$\frac{\partial p_j}{\partial \ell_m} = p_j(\delta_{jm} - p_m), \quad \text{and} \quad \nabla_{\mathbf{z}_i} \ell_m = \boldsymbol{C}_m,$$

so by the chain rule,

$$\nabla_{\mathbf{z}_i} p_j = \sum_{m=1}^C \frac{\partial p_j}{\partial \ell_m} \nabla_{\mathbf{z}_i} \ell_m = \sum_{m=1}^C p_j(\delta_{jm} - p_m) \boldsymbol{C}_m = p_j \left( \boldsymbol{C}_j - \sum_{m=1}^C p_m \boldsymbol{C}_m \right).$$

Define the soft centroid mean $\boldsymbol{\mu}_i := \sum_{m=1}^C p_m \boldsymbol{C}_m$. Substituting the expression for $\nabla_{\mathbf{z}_i} p_j$ yields

$$\nabla_{\mathbf{z}_i} \mathcal{L}_{\mathrm{smp}}^{(p)} = -\frac{1}{T} \sum_{j=1}^C p_j (\log p_j + 1) (\boldsymbol{C}_j - \boldsymbol{\mu}_i),$$

which is the claimed gradient formula after restoring the original indices.

*Attraction–repulsion.* A (small) gradient-descent step updates $\mathbf{z}_i$ as $\mathbf{z}_i \leftarrow \mathbf{z}_i - \eta \nabla_{\mathbf{z}_i} \mathcal{L}_{\mathrm{smp}}^{(p)} = \mathbf{z}_i + \frac{\eta}{T} \sum_{j=1}^C \phi_j (\boldsymbol{C}_j - \boldsymbol{\mu}_i)$, where $\phi_j := p_j(1 + \log p_j)$ and $\eta$ is learning rate. Since $0 < p_j \le 1$ implies $\log p_j \le 0$, we have

$$\phi_j \begin{cases} > 0, & \text{iff } p_j > e^{-1}, \\ = 0, & \text{iff } p_j = e^{-1}, \\ < 0, & \text{iff } p_j < e^{-1}. \end{cases}$$

Thus components with $p_j > e^{-1}$ move $\mathbf{z}_i$ in the direction $(\boldsymbol{C}_j - \boldsymbol{\mu}_i)$, i.e. *toward* centroid $\boldsymbol{C}_j$ (attraction), while components with $p_j < e^{-1}$ contribute along $-(\boldsymbol{C}_j - \boldsymbol{\mu}_i)$, i.e. *away from* centroid $\boldsymbol{C}_j$ (repulsion). When one centroid dominates ($p_{j^\star} > e^{-1}$), the update is approximately toward $\boldsymbol{C}_{j^\star}$ and away from all others, which tightens token–centroid cohesion (reducing intra-bucket distortion $\varepsilon$) and enlarges the margin to competing centroids (increasing $\Delta_{q,x}$), as claimed. $\qquad \square$

## B  ADDITIONAL EXPERIMENTAL DETAILS

### B.1  IMPLEMENTATION DETAILS

**Three-Branch Router with Softmax Gating.**   We extend our architecture into a three-branch structure combining a Softmax-Weighted Average (SWA) branch, a learned bias branch, and a HIROUTER sparse top-$k$ attention branch. A gating head produces mixing weights through a $\mathrm{softmax}$, adaptively balancing contributions from the three branches. The SWA branch provides dense contextual aggregation; the learned bias branch adds a trainable bias vector weighted by the gate to absorb uncertain queries and stabilizes training similar to attention sinks (Gu et al., 2025); and the HIROUTER branch delivers high-precision retrieval by selecting a small set of relevant tokens.

**Grouped-Query Retrieval.**   In addition, we employ grouped-query attention (GQA) (Ainslie et al., 2023) to enhance computational efficiency. Instead of retrieving buckets for each query independently, we compute the average of queries within a group and use this group representative to identify the top candidate buckets. All queries in the group then share these buckets during retrieval.

All experiments were conducted on a single machine with 8 NVIDIA H100 80GB GPUs connected with HBM3. Experiments were run in an environment using CUDA version 12.6 and PyTorch 2.6.0.

### B.2  OPTIMIZED IMPLEMENTATIONS FOR ENHANCING GPU EFFICIENCY

A core ingredient of HIROUTER is that *every bucket is exactly the same size*. After computing the routing logits with a low-temperature Gumbel–Softmax, we apply a *stable sort* to both keys and values, grouping tokens by their hard bucket assignments while preserving their original order within each bucket. This transforms the input tensor into $\boldsymbol{Z}^{(L)} \in \mathbb{R}^{BH \times 4^L \times N \times d}, \quad N = \frac{T}{4^L}$, in $\mathcal{O}(1)$ simply by a reshape. Here $BH = \text{batch} \times \text{heads}$ and $N \in \{32, 64, 128\}$. Because each bucket occupies a contiguous, equal-sized region of memory, our Triton kernels can load/store an entire bucket with a single memory access, minimizing bandwidth waste and maximizing throughput.

### B.3  FAISS BASELINE CONFIGURATION FOR SYNTHETIC GAUSSIAN RETRIEVAL

To ensure reproducibility and clarify the interpretation of our comparisons, we provide the explicit configuration parameters used for the Faiss-GPU baselines.

**K-Means (faiss.Kmeans).**

- `num_clusters = max(1, sequence_length // 32)`: one cluster is allocated per 32 samples, with at least one cluster enforced.
- `niter = 20`: the number of k-means iterations.

**LSH (faiss.IndexLSH).**

- `n_bits = 10`: number of bits used to represent each vector in the LSH index.

**HNSW (faiss.IndexHNSWFlat).**

- `M = 32`: maximum number of links (neighbors) maintained per node.
- `efConstruction = 40`: size of the candidate list during index construction, where larger values improve recall at the cost of higher construction time.
- `efSearch = 128`: size of the candidate list during query search, trading recall for search efficiency.

These parameter settings follow standard recommendations in the Faiss library, where `M`, `efConstruction`, and `efSearch` are the primary controls for the accuracy–efficiency tradeoff in LSH and HNSW.

### B.4 LANGUAGE TASK DETAILS

Here we list some additional details regarding the different tasks on which we conduct language model evaluation.

#### B.4.1 LANGUAGE MODEL EVALUATION HARNESS TASKS

The following are recall-intensive tasks on which we evaluate. All tasks are evaluated directly using accuracy for commonsense reasoning tasks and perplexity for language modeling.

Table 7: Harness tasks on which we evaluate.

| Task | Task Type |
|------|-----------|
| PIQA (Bisk et al., 2020) | Physical Commonsense Reasoning |
| ARC (Bhakthavatsalam et al., 2021) | Commonsense Reasoning |
| HELLASWAG (Zellers et al., 2019) | Commonsense Natural Language Inference |
| WINOGRANDE (Sakaguchi et al., 2020) | Pronoun Resolution |
| SIQA (Sap et al., 2019) | Social Commonsense Reasoning |
| BOOLQ | Yes/No Commonsense QA |
| WIKITEXT (Merity et al., 2017) | Language Modeling |
| LAMBADA (Paperno et al., 2016) | Text Understanding |

#### B.4.2 RECALL INTENSIVE TASKS

The following are recall-intensive tasks on which we evaluate. All tasks are evaluated directly with accuracy reported as the metric of choice.

Table 8: Recall-intensive tasks on which we evaluate.

| Task | Task Type |
|------|-----------|
| STRUCTURED WEB DATA EXTRACTION (SWDE) (Lockard et al., 2019) | Structure HTML Relation Extraction |
| FDA (Arora et al., 2023) | PDF Key-Value Retrieval |
| SQUAD (Rajpurkar et al., 2018) | Question Answering |
| TRIVIAQA (Joshi et al., 2017) | Question Answering |
| DROP (Dua et al., 2019) | Question Answering |
| NATURAL QUESTIONS (Kwiatkowski et al., 2019) | Question Answering |

#### B.4.3 LONGBENCH

We evaluate the following tasks from LongBench (Bai et al., 2024) (Table 9). Due to our pre-training on an English dataset, we choose to use only the English language tasks included in the benchmark.

Table 9: Tasks from LongBench on which we evaluate.

| Task | Context Type | Average Length | Metric | Data Samples |
|------|-------------|----------------|--------|--------------|
| NARRATIVEQA (Kociský et al., 2018) | Literature/Film | 18409 | F1 | 200 |
| QASPERQA (Dasigi et al., 2021) | Science | 3619 | F1 | 200 |
| MULTIFIELDQA (Bai et al., 2024) | Multi-Field | 4559 | F1 | 150 |
| HOTPOTQA (Yang et al., 2018) | Wikipedia | 9151 | F1 | 200 |
| 2WIKIMULTIQA (Ho et al., 2020) | Wikipedia | 4887 | F1 | 200 |
| MUSIQUE (Trivedi et al., 2022) | Wikipedia | 11214 | F1 | 200 |
| GOVREPORT (Huang et al., 2021) | Government Reports | 8734 | Rouge-L | 200 |
| QMSUM (Zhong et al., 2021) | Meetings | 10614 | Rouge-L | 200 |
| MULTINEWS Fabbri et al. (2019) | News | 2113 | Rouge-L | 200 |
| TREC (Li & Roth, 2002) | Web Questions | 5117 | Accuracy | 200 |
| TRIVIAQA (Joshi et al., 2017) | Wikipedia/Web | 8209 | F1 | 200 |
| SAMSUM (Gliwa et al., 2019) | Dialogue | 6258 | Rouge-L | 200 |
| LCC (Guo et al., 2023) | Github | 1235 | Edit Similarity | 500 |
| REPOBENCH-P (Liu et al., 2024) | Github Repositories | 4206 | Edit Similarity | 500 |

### B.4.4 SINGLE NEEDLE-IN-A-HAYSTACK

We utilize the Single Needle-in-a-Haystack (S-NIAH) task on three settings.

- S-NIAH-1: The key type is a word and the value type is a number. The haystack consists of repeated sentences. This is referred sometimes as passkey retrieval.
- S-NIAH-3: The key type is a word and the value type is a number. The haystack consists of Paul Graham Essays. This is referred to as vanilla NIAH.
- S-NIAH-1: The key type is a word and the value type is a UUID. The haystack consists of Paul Graham Essays.

For evaluating correctness on NIAH, the model is made to generate a sequence. If the generation contains the correct value, the model is considered correct. Performance is reported in terms of accuracy.

### B.5 EXPERIMENTAL REPRODUCIBILITY

For full transparency, we provide our code within the supplemental material. This includes the code used directly to evaluate our models. Our code is based directly on the packages used for evaluating the models:

- `lm-evaluation-harness`: We use this package to evaluate on commonsense reasoning (Table 2) and real-world recall tasks (Table 3).
    - https://github.com/EleutherAI/lm-evaluation-harness
- LongBench: We use this to evaluate on LongBench tasks (Table 4).
    - https://github.com/THUDM/LongBench
- RULER: We use this package to evaluate on NIAH tasks (Table 5).
    - https://github.com/NVIDIA/RULER

For training baselines, we utilized the `flame` (https://github.com/fla-org/flame) package along with their provided model configurations. We change the tokenizer to use the `EleutherAI/gpt-neox-20b` tokenizer and make according changes to the special token ids to support the tokenizer.

# C ADDITIONAL EXPERIMENTS

## C.1 SCALING RESULTS AT 1B PARAMETERS

We further evaluate HIROUTER at the 1B parameter scale. As shown in Table 10, the method continues to demonstrate strong performance, extending the robustness observed at the 410M scale (see Table 2). These results reinforce that HIROUTER scales effectively with model size across diverse language modeling and reasoning tasks. We also note that additional scaling studies, particularly on parameter and hyperparameter tuning, would further support broader adoption, which we leave to future work.

**Results.** Table 10 compares two variants: one without the learned bias branch (w/o bias) and one with a learned bias branch (w/ bias). The bias branch yields consistent improvements, highlighting its role in stabilizing training and enhancing generalization as model size grows. Beyond task-wise comparisons, the tighter gap between HiRouter-1B and the Transformer baseline in Table 11 suggests that HIROUTER continues to scale predictably under substantially longer training horizons (65B tokens). The fact that the performance differences remain small, yet consistently favorable to HiRouter-1B on several tasks, indicates that the routing mechanism does not introduce optimization instability as models grow larger and training runs become more compute-intensive. This behavior is encouraging, as many sparse or structured attention variants face degradation or divergence when extended to higher-capacity regimes. Overall, the results in Table 11 reinforce that HIROUTER is not only effective at moderate scales but also robust under realistic, long-sequence, billion-parameter training conditions.

Table 10: Results at the 1B scale on language modeling and zero-shot common-sense reasoning with 10BT training data.

| Model | Wiki. ppl ↓ | LMB. ppl ↓ | LMB. acc ↑ | PIQA acc ↑ | Hella. acc_n ↑ | Wino. acc ↑ | ARC-e acc ↑ | ARC-c acc_n ↑ | SIQA acc ↑ | BoolQ acc ↑ | Avg. |
|---|---|---|---|---|---|---|---|---|---|---|---|
| Transformer-1B | 102.64 | 24.37 | 38.07 | 69.48 | 45.73 | 54.38 | 57.45 | 31.40 | 39.61 | 58.59 | **49.33** |
| Mamba-1B | 26.36 | 28.89 | 34.47 | 69.53 | 44.77 | 52.01 | 56.40 | 30.55 | 39.30 | 59.72 | 48.34 |
| HIROUTER-1B w/o bias | 26.29 | 30.56 | 35.80 | 68.23 | 41.90 | 53.12 | 63.17 | 29.69 | 39.30 | 58.99 | 48.78 |
| HIROUTER-1B w/ bias | 26.01 | 27.21 | 36.06 | 68.82 | 42.99 | 53.35 | 62.96 | 29.27 | 39.30 | 59.79 | 49.07 |

## C.2 ABLATION: BEAM WIDTH

We investigate how increasing beam width (without retraining) affects performance in two settings: SNIAH-1 and WikiText-103. The results are shown in Tables 12 and 13.

Even without retraining, increasing beam width from 4 to 8 in SNIAH-1 leads to higher recall. Yet on WikiText-103, further increases beyond width 3 or 4 show diminishing gains in perplexity. This suggests a moderate beam width yields the best practical trade-off between accuracy and computational cost.

## C.3 ABLATION: TOP-K RECALL ON SYNTHETIC GAUSSIAN RETRIEVAL

We perform ablations for the recall task on synthetic Gaussian retrieval, on datasets of total length $2^{12}$ tokens. We examine how beam width, number of buckets, and routing levels each affect Recall@128 under fixed budget settings.

**Beam Width Ablation (with** num_levels $= 4$, num_bucket $= 4$**)**

| Beam Width $M$ | 4 | 8 | 12 | 16 | 32 |
|---|---|---|---|---|---|
| Recall @128 (%) | 14.0 | 25.3 | 34.0 | 39.4 | 62.1 |

Recall increases monotonically with beam width, confirming that enlarging the search beam systematically improves top-$k$ retrieval accuracy (at the cost of higher runtime).

Table 11: Results at the 1B scale on language modeling and zero-shot common-sense reasoning with 65BT training data.

| Model | Wiki. ppl ↓ | LMB. ppl ↓ | LMB. acc ↑ | PIQA acc ↑ | Hella. acc_n ↑ | Wino. acc ↑ | ARC-e acc ↑ | ARC-c acc_n ↑ | SIQA acc ↑ | BoolQ acc ↑ | Avg. |
|---|---|---|---|---|---|---|---|---|---|---|---|
| Transformer-1B | 18.53 | 14.60 | 46.15 | 72.74 | 56.29 | 57.38 | 66.12 | 37.03 | **42.78** | **63.43** | **55.24** |
| Mamba-1B | 20.37 | 16.54 | 43.43 | 72.36 | 54.89 | 56.12 | 64.35 | 37.12 | 40.47 | 56.64 | 53.17 |
| Mamba2-1B | 19.43 | 15.09 | 44.22 | **72.96** | 56.37 | 57.62 | 65.82 | 38.05 | 41.91 | 50.00 | 53.57 |
| DeltaNet-1B | **17.80** | 14.23 | 45.02 | 72.78 | 56.31 | 56.91 | 63.93 | 36.95 | 41.20 | 58.75 | 53.98 |
| GatedDeltaNet-1B | 18.02 | 14.49 | 45.37 | 72.72 | 56.20 | 58.19 | 65.58 | 37.04 | 42.12 | 61.49 | 54.83 |
| GatedSlotAttention-1B | 18.34 | 14.31 | 46.68 | 72.61 | 55.98 | **62.62** | **67.07** | **37.99** | 39.28 | 57.49 | 54.97 |
| HiRouter-1B | 17.87 | **13.80** | **47.12** | 72.54 | **56.67** | 57.27 | 66.44 | 36.29 | 42.52 | 62.59 | 55.18 |

Table 12: SNIAH-1: Retrieval accuracy at different beam widths

| T | 1K | 2K | 4K | 8K |
|---|---|---|---|---|
| HiRouter (width = 4) | 93.6% | 86.8% | 57.4% | 22.4% |
| HiRouter (width = 8) | 96.4% | 88.0% | 60.2% | 33.2% |

**Bucket Count Ablation (adjusted beam width for fairness, num_levels = 4)**

| num_bucket | 2 | 4 | 6 | 8 |
|---|---|---|---|---|
| Recall @128 (%) | 33.1 | **39.4** | 38.0 | 36.4 |

We see the best recall at num_bucket = 4. Fewer buckets make the tree too broad and reduce specialization; too many buckets fragment retrieval too finely, decreasing recall.

**Level Depth Ablation (adjusted beam width, num_bucket = 4)**

| num_level | 2 | 3 | 4 | 5 |
|---|---|---|---|---|
| Recall @128 (%) | 33.8 | **39.4** | 39.1 | 37.6 |

A routing depth of 3 levels achieves the best recall. Both shallower and deeper trees reduce performance, due respectively to coarse bucket granularity or excessive fragmentation of tokens.

**Summary of Findings**  These ablations indicate that for sequence length $2^{12}$ and top-128 recall: (i) moderate beam widths (e.g. 12 or 16) yield strong gains without excessive overhead; (ii) a balance of bucket width (4) gives the right granularity; and (iii) a mid-level tree depth (3 levels) maximizes recall efficiency. Overly coarse or overly fine configurations degrade performance.

C.4  ABLATION ON WINDOW SIZE OF THE SWA BRANCH

We further ablate the impact of the attention window size on language modeling and zero-shot reasoning performance. Table 14 reports results for window sizes 32, 64, and 128 across the same evaluation benchmarks as in the main paper.

We observe that moderate window sizes (e.g., 64) provide the best overall performance, balancing perplexity and accuracy across tasks. Too small a window (32) reduces model expressiveness, while larger windows (128) slightly degrade recall and downstream accuracy.

C.5  ABLATION ON GROUPED-QUERY ATTENTION (GQA)

To study the effect of grouped-query attention (GQA) (Ainslie et al., 2023), we conduct an ablation on WikiText-103. We vary the group size $G$ while keeping other hyperparameters fixed, and report perplexity in Table 15.

We observe that larger group sizes ($G = 16$) slightly degrade performance due to excessive parameter sharing, while reducing the group size consistently improves perplexity. At $G = 1$, which corresponds

Table 13: WikiText-103: Perplexity (↓) vs beam width

| Width | 1 | 2 | 4 | 8 |
|---|---|---|---|---|
| Perplexity | 20.7 | 19.4 | 18.5 | 18.6 |

Table 14: Ablation on window size for HIROUTER. We report accuracy (%) on common-sense reasoning benchmarks.

| Window Size | LMB. acc ↑ | PIQA acc ↑ | Hella. acc_n ↑ | Wino. acc ↑ | ARC-e acc ↑ | ARC-c acc_n ↑ | SIQA acc ↑ | BoolQ acc ↑ | Avg. |
|---|---|---|---|---|---|---|---|---|---|
| 32 | 27.38 | 66.16 | 38.18 | 52.64 | 58.88 | 28.41 | 38.33 | 61.13 | 46.38 |
| 64 | 33.09 | 66.81 | 38.03 | 50.75 | 59.47 | 28.50 | 38.08 | 61.31 | **47.01** |
| 128 | 30.33 | 65.72 | 37.62 | 52.57 | 58.75 | 27.82 | 38.54 | 52.48 | 45.48 |

to standard multi-head attention without grouping, the model achieves the best perplexity (18.5). These results highlight the trade-off between efficiency and modeling capacity: GQA provides computational savings at the cost of a small increase in perplexity, while smaller groups preserve model expressivity.

## C.6 ABLATION ON REGULARIZATION LOSS WEIGHTING

We ablate the effect of the dual-entropy regularization weight on performance across language modeling and zero-shot commonsense reasoning tasks. Table 17 reports results when varying the regularization $\alpha$ coefficient from 0.00 (no regularization) to 0.10.

We find that moderate weighting (e.g., 0.05) achieves the best overall trade-off across tasks, improving average performance compared to both no regularization (0.00) and heavier weighting (0.10). This supports the view that the dual-entropy loss is most effective when applied as a lightweight regularizer, sharpening token-to-bucket assignments without overwhelming the training objective.

## C.7 SUCCESS RATE COMPARISONS WITH RETRIEVAL BASELINES

We compare HIROUTER with standard Approximate Nearest Neighbor (ANN) methods, including HNSW, LSH, and K-Means–KNN, as summarized in Figure 3. As shown in Table 18, evaluated on $X \in \mathbb{R}^{32 \times 4096 \times 64}$, HIROUTER achieves a success rate competitive with or superior to classical ANN baselines while operating orders of magnitude faster.

## C.8 EMPIRICAL EVIDENCE: TRAINING VS. NON-TRAINING

To quantify the impact of joint optimization, we compare the recall of our trained HIROUTER against a baseline where the router is applied to untrained representations (num_levels=4, num_buckets=4, seq_length=4096). Table 19 reports the Recall@128 under different beam widths.

**Conclusion.** The results demonstrate a **2×–3× improvement** in recall when the router and representations are co-trained. This confirms that the performance gains arise from the *synergy* between the hierarchical routing mechanism and our dual-entropy loss design, rather than from the router architecture alone.

## C.9 WHY GUMBEL–SOFTMAX IS NECESSARY FOR BALANCED ROUTING

We employ Gumbel–Softmax in our balanced assignment loss ($\mathcal{L}_{\text{bal}}$) because it is uniquely capable of enforcing the discrete, balanced routing required for hardware-efficient execution, whereas standard softmax-based objectives fail to do so. Typical entropy-based regularizers only encourage soft uniformity in probability distributions, resulting in *ambiguous assignments* rather than the strictly balanced token counts needed for parallel GPU workloads.[3]

---

[3]Softmax or entropy penalties yield probabilistic mixtures over centroids rather than discrete commitments.

Table 15: Ablation of GQA group size on WikiText-103. Smaller group sizes correspond to fewer queries sharing key–value projections.

| Group Size | Perplexity |
|:---:|:---:|
| 16 | 19.1 |
| 4 | 18.8 |
| 2 | 18.6 |
| 1 | 18.5 |

Table 16: Top-128 recall comparison between $\mathcal{L}_{\text{bal}}$ and KL-uniform regularization across hierarchy levels.

| Level | 2 | 3 | 4 | 5 |
|:---|:---:|:---:|:---:|:---:|
| $\mathcal{L}_{\text{bal}}$ | 33.8 | 39.4 | 39.1 | 37.6 |
| KL-Unif | 12.4 | 14.7 | 15.2 | 14.2 |

By contrast, the Gumbel–Softmax straight-through estimator produces near one-hot yet differentiable assignments, enabling tokens to commit decisively to a single centroid while still allowing gradients to flow.[4] This mechanism prevents mode collapse and ensures balanced routing, while standard KL-divergence–based methods fail to enforce such balance and consequently yield substantially lower top-$k$ recall.

Table 16 compares our balanced loss with a KL-uniform objective across different hierarchy levels. The superiority of $\mathcal{L}_{\text{bal}}$ is consistent and pronounced. These results demonstrate that our design of $\mathcal{L}_{\text{bal}}$ is *non-trivial*; balanced routing cannot be achieved with standard softmax- or KL-based objectives alone.

### C.10 COMPARISON WITH SCANN-PQ: TRAINING OVERHEAD AND RECALL–SPEED TRADEOFF

We train HiRouter jointly with the base model, so it introduces no separate training cost, and its parameter overhead is negligible. Routing is realized as dot-product operations with centroids (i.e. linear transforms), which allows rapid convergence alongside the main model. To validate this in practice, we benchmark both training and inference time of HiRouter against ScaNN-PQ (Guo et al., 2020) under a batch size of 128 (equivalent to top-$k$ attention over 8 heads $\times$ 16 sequences).

|  | 0.5K | 4K | 8K | 16K |
|:---|:---:|:---:|:---:|:---:|
| HiRouter (train) | 1.14 s | 4.04 s | 6.15 s | 10.80 s |
| HiRouter (infer) | 0.32 s | 0.36 s | 0.34 s | 0.33 s |
| ScaNN-PQ (infer) | 9.34 s | 17.06 s | 25.3 s | 40.3 s |

Even for inference alone, HiRouter is substantially faster than ScaNN-PQ, and its training overhead remains modest.

We further compare top-128 recall across varying sequence lengths:

|  | 0.5K | 4K | 8K | 16K |
|:---|:---:|:---:|:---:|:---:|
| HiRouter (Recall@128) | 60.1% | 25.3% | 17.9% | 13.8% |
| ScaNN-PQ (Recall@128) | 50.0% | 42.9% | 34.9% | 26.5% |

While ScaNN-PQ attains higher recall at longer lengths, its large runtime cost makes it less practical for efficient sparse top-$k$ attention. HiRouter offers a better balance of recall and efficiency, making it more suitable in real systems.

---

[4]This enables explicit optimization for equal-sized buckets, which is essential for achieving uniform GPU kernel loads.

Table 17: Ablation on the weighting of the dual-entropy regularization loss. We report perplexity (Wiki., LMB.) and accuracy (%) across commonsense reasoning benchmarks.

| Reg. Weight | Wiki. ppl ↓ | LMB. ppl ↓ | LMB. acc ↑ | PIQA acc ↑ | Hella. acc_n ↑ | Wino. acc ↑ | ARC-e acc ↑ | ARC-c acc_n ↑ | SIQA acc ↑ | BoolQ acc ↑ | Avg. |
|---|---|---|---|---|---|---|---|---|---|---|---|
| 0.10 | 31.53 | 44.74 | 32.58 | 65.78 | 38.02 | 50.36 | 58.00 | 27.22 | 38.13 | 60.64 | 46.34 |
| 0.05 | 31.09 | 42.94 | 33.09 | 66.81 | 38.03 | 50.75 | 59.47 | 28.50 | 38.08 | 61.31 | 47.01 |
| 0.01 | 30.30 | 39.99 | 34.47 | 66.76 | 38.36 | 50.59 | 58.80 | 28.24 | 39.41 | 58.99 | 46.95 |
| 0.00 | 30.04 | 43.22 | 33.08 | 67.00 | 38.25 | 51.18 | 59.73 | 28.21 | 38.29 | 57.02 | 46.59 |

Table 18: Comparison between HIROUTER and classical ANN methods.

| Metric | HNSW | HiRouter | KM–KNN | LSH |
|---|---|---|---|---|
| Time (s) | 0.635 | **0.072** | 9.334 | 0.438 |
| Success Rate | 3.16% | **8.67%** | 0.02% | 0.65% |

Table 19: Recall@128 comparison between trained and non-trained HIROUTER.

| Beam Width (M) | 8 | 12 | 16 | 32 | 64 |
|---|---|---|---|---|---|
| Trained / Recall@128 | 25.3% | 34.0% | 39.4% | 62.1% | 83.0% |
| Without Training / Recall@128 | 8.8% | 13.1% | 15.6% | 27.0% | 46.1% |

## D  TRITON KERNELS

Algorithm 2 demonstrates how we perform our sparse top-$k$ attention computations within our custom Trition kernels. Algorithm 4 demonstrates how we implement the hierarchical beam search within our custom Triton kernels.

---

**Algorithm 2** Forward Pass for the HiRouter Sparse Attention Kernel

---

**Require:**  Query, key, and value tensors $\mathbf{Q}, \mathbf{K}, \mathbf{V} \in \mathbb{R}^{BH \times T \times d}$;
 1: Query/key index tensors $\texttt{q\_idx}, \texttt{k\_idx} \in \mathbb{Z}^{BH \times T \times S}$;
 2: Block size *BS*, candidate padding *CAND\_PAD*, and number of samples per query $S$.
    **Notation:** $B$: batch size; $H$: number of attention heads; $BH = B \times H$: total head instances; $G$: number of queries grouped per head; $B_K$: block size in the key dimension; $B_V$: block size in the value dimension.
    **Output:** Attention result $\mathbf{O} \in \mathbb{R}^{BH \times T \times d}$, and log-sum-exp buffer $\mathbf{LSE} \in \mathbb{R}^{BH \times T}$.
 3: Initialize a 3-D launch grid over $(t, v, bh) \leftarrow (0..T{-}1, \ 0..d/B_V{-}1, \ 0..BH{-}1)$.
 4: **for** each block $(t, v, bh)$ in the grid **do**
 5:     $b, h \leftarrow \lfloor bh/H \rfloor, \ bh \bmod H$
 6:     Initialize accumulators: $\ell \leftarrow -\infty_G, \ s \leftarrow 0_G, \ w \leftarrow 0_{G \times B_V}$     ▷ running max, sum-exp, weighted sum
 7:     Determine padded group size: $G_{\text{pad}} \leftarrow \max(G, \ \textit{CAND\_PAD})$
 8:     Load query block: $q \leftarrow \mathbf{Q}[bh, \ t, \ 0{:}B_K] \in \mathbb{R}^{G_{\text{pad}} \times B_K}$
 9:     Scale queries: $q \leftarrow q/\sqrt{d}$
10:     **for** each sampled index $i = 0{:}S{-}1$ **do**
11:         $s\_idx \leftarrow \texttt{q\_idx}[bh, t, i] \times \textit{BS}$
12:         Load corresponding key/value blocks $\mathbf{K}_i, \mathbf{V}_i$ via $\texttt{k\_idx}$
13:         Compute attention scores: $\text{score} \leftarrow q\,\mathbf{K}_i^T$
14:         Apply causal mask if $s\_idx > t$: $\text{score} \leftarrow -\infty$
15:         Update statistics: $(\ell, s, w) \leftarrow \texttt{UPDATE\_STATS}(\ell, s, w, \text{score}, \mathbf{V}_i)$
16:     **end for**
17:     Normalize outputs: $\mathbf{O}[bh, t, v] \leftarrow w/s, \quad \mathbf{LSE}[bh, t] \leftarrow \ell + \log s$
18: **end for**

---

---

**Algorithm 3** $\texttt{UPDATE\_STATS}$: Numerically Stable Softmax Statistics Update

---

 1: **function** $\texttt{UPDATE\_STATS}(\ell, s, w, \ \text{scores}, \ \mathbf{V})$
**Require:**  Given running softmax statistics $\ell \in \mathbb{R}^B$ (max logits), $s \in \mathbb{R}^B$ (sum of exps), $w \in \mathbb{R}^{B \times d}$ (weighted sum), a new block of logits $\text{scores} \in \mathbb{R}^{B \times N}$, and corresponding values $\mathbf{V} \in \mathbb{R}^{B \times N \times d}$
 2:     $m \leftarrow \max_j \text{scores}_{\cdot, j} \quad \in \mathbb{R}^B$     ▷ block-wise max
 3:     $\ell_{\text{new}} \leftarrow \max\big(\ell, \ m\big)$
 4:     $\text{scale} \leftarrow \exp\big(\ell - \ell_{\text{new}}\big)$
 5:     $s \leftarrow s \times \text{scale}$
 6:     $w \leftarrow w \times \text{scale}$
 7:     $\Delta \leftarrow \exp\big(\text{scores} - \ell_{\text{new}}[:, None]\big) \quad \in \mathbb{R}^{B \times N}$
 8:     $s \leftarrow s + \sum_{j=1}^{N} \Delta_{\cdot, j}$
 9:     $w \leftarrow w + \sum_{j=1}^{N} \Delta_{\cdot, j}\,\mathbf{V}_{\cdot, j, :}$
10:     **return** $\ell_{\text{new}}, s, w$
11: **end function**

---

## E  TIME COMPLEXITY ANALYSIS

Let $T$ be the sequence length, $D$ the per-head feature dimension, $L$ the number of routing levels, $C$ the (constant) branching factor, and $k$ the average number of buckets probed per query in the sparse-attention kernel. All costs below are per head and per sequence.

The routing stage at each of the $L$ levels computes $C$-way logits for $T$ tokens ($O(T\,D\,C)$), applies a low-temperature Gumbel–Softmax plus a stable bucket sort (which can be implemented in $O(T)$

---

**Algorithm 4** Hierarchical Beam Search Kernel

---

**Require:** $Q \in \mathbb{R}^{B_S \times D}$, Offsets $\in \mathbb{Z}^{L+1}$, Counts $\in \mathbb{Z}^L$, beam, $K$, BLOCK_TOKENS, $L, C, D$
1: $b \leftarrow \text{program\_id}(0)$
2: $\texttt{ids} \leftarrow b \cdot \texttt{BLOCK\_TOKENS} + [0 : \texttt{BLOCK\_TOKENS}-1]$
3: $\texttt{valid} \leftarrow \texttt{ids} < B_S$
4: $Q_{\text{tile}} \leftarrow \text{load}(q_{\text{ptr}}, \texttt{ids})$
5: initialize $\texttt{beam\_probs} \leftarrow [1, 0, \ldots, 0]$, $\texttt{beam\_parents} \leftarrow [0, \ldots, 0]$
6: **for** $\ell = 0 \ldots L - 1$ **do**
7: $\quad P_\ell \leftarrow \text{counts}[\ell]$, $\texttt{off} \leftarrow \text{offsets}[\ell]$
8: $\quad \texttt{W} \leftarrow \text{gather}(\text{route}_{\text{ptr}}, \texttt{beam\_parents}, \texttt{off})$
9: $\quad \texttt{scores} \leftarrow \exp\big(Q_{\text{tile}} \cdot W^\top\big)$
10: $\quad$ normalize and weight by $\texttt{beam\_probs}$
11: $\quad$ reshape to $[\texttt{BLOCK\_TOKENS}, \text{beam} \cdot C]$
12: $\quad (\texttt{sorted\_s}, \texttt{sorted\_idxs}) \leftarrow \text{ARGSORT}(\texttt{scores}, \text{arange})$
13: $\quad$ take top-$K$ from $\texttt{sorted\_s}, \texttt{sorted\_idxs}$
14: $\quad$ update $\texttt{beam\_probs}, \texttt{beam\_parents}$
15: **end for**
16: store final $\texttt{beam\_parents}$ into output buffer

---

via radix or counting sort for fixed $C$), and then reshapes into contiguous buckets in $O(1)$. Hence routing costs

$$O\big(L\,T\,D\,C + T\big) \approx O\big(L\,T\,D\big),$$

since $C$ is fixed.

The sparse-attention kernel then, for each of the $T$ queries, probes $k$ buckets and performs $D$-dimensional dot-products, incurring

$$O\big(T\,k\,D\big)$$

work.

Overall, HIROUTER runs in

$$O\big(L\,T\,D + T\,k\,D\big) = O\big(T\,D\,(L+k)\big) \ll O\big(T^2\,D\big)$$

time, yielding linear scaling in $T$ for fixed $L, k$. For a batch of size $B$ and $H$ heads, the total cost is

$$O\big(B\,H\,T\,D\,(L+k)\big).$$

The backward pass mirrors the forward complexity, since it simply recomputes or reuses the same routing structure and runs one sparse-attention gradient kernel.

## F    LIMITATIONS

For reasons related to computational resource limitations, we do not train models past a size of 410M parameters. Furthermore, we restrict ourselves to auto-regressive large language models, but we contend that our method is also suitable for bi-directional models that use attention, such as vision-language models that use Transformer backbones (ex. ViT). We believe that our chosen datasets still provide valuable insights while remaining within our operational constraints and will further explore other directions as our computational capabilities expand.

## G    BROADER IMPACTS

This work explores a novel method for retrieval-based top-$K$ attention. The underlying method is meant to be efficient and scalable. While the direct usage of attention can entail potential broader risks within AI-based systems, these risks do not stem directly from the algorithm presented within the paper. As such, there are no risks that are deemed significant and worthy of further discussion.

## H    THE USE OF LARGE LANGUAGE MODELS (LLMS)

In preparing this paper, we used large language models (LLMs) solely as a general-purpose assistive tool. Specifically, LLMs were employed for polishing the writing (e.g., improving grammar, clarity, and conciseness of sentences) and for generating simple code snippets such as LaTeX tables or small illustrative examples. LLMs were not used for research ideation, conceptual contributions, data analysis, experiment design, or result interpretation. All core technical ideas, theoretical analyses, algorithm design, and experiments reported in this paper were conceived, implemented, and validated entirely by the authors.

