# OpenReview forum: "Hierarchical Routers for Efficient Top-k Retrieval in Sparse Attention"
_ICLR.cc/2026/Conference — Submitted to ICLR 2026_

### Official Review · Reviewer_jZmf · 2025-10-27

**Soundness:** 3
**Presentation:** 2
**Contribution:** 3
**Rating:** 6
**Confidence:** 4

**Summary:**

The paper proposes a trainable hierarchical routing scheme for implementing top-k attention, observing performance improvements in Transformers without sacrificing model quality.

**Strengths:**

1. The paper tackles an important problem - making Transformers resource efficient while not sacrificing quality. Solutions in this space are undoubtedly influential and impactful in the AI space.
2. The paper proposes a novel mechanism for implementing top-k attention:
    * Each query token is routed throughout a hierarchical tree structure that allows for the computation of the top-k tokens.
    * Routing is performed by maintaining new weight "centroid" matrices and multiplying the query with those matrices. This results in an empirical distribution over the children of a node in the tree. The distribution is smoothed via the softmax Gumbel trick and the two highest probability buckets are selected. In the end, all the candidate key tokens are selected in the leaves of the tree.
    * Sparse attention is then calculated based on these candidates.
    * To encourage separability of the centroid clusters as well as fair partitioning the authors propose to use an entropy-like regularizer.
Though the ideas in this mechanism are not new to this paper (more details below), their combination in this context definitely constitutes an important contribution.
3. The paper performs a wide variety of experiments to verify that the proposed method works as advertised. The experiments are done on empirical datasets as well as on small and larger Transformers. The results show that in general the proposed mechanism does achieve more efficient inference for large context lengths while not degrading the model's quality.
4. Theoretical insights are also a positive for this paper. They may not actively show that the learning process distills helpful information about the token's structure, but they support the intuition behind defining entropy-based regularizers.
5. Potential for parallelizability and GPU utilization is also a very important plus for this method.

**Weaknesses:**

1. I think the paper's main method is not presented as well as it could be. Specifically, Section 3.2 can definitely benefit from a rewrite to make the hierarchical routing method more explicit and easy to parse through. Without Figure 2, for example, it would have been very difficult for me to understand what is actually happening. I think the correct way to present this technique is via an explicit algorithm that lists the inputs, expected outputs and hyperparameters. Then statements about the runtime of the algorithm (Appendix E) can be made in the context of such an algorithm.
2. I think the paper can situate itself better in the context of prior work. Many of the ideas it builds upon were introduced in previous works, which often provide more theoretical justification for why these methods work well with attention. I think such context is especially important for papers doing work in such a crowded research space.
    * Clustering-based algorithms for efficient attention and KV cache compression: See, for example, the work of [1]: they maintain representative clusters of the token embeddings in similar fashion as this paper. Also, the work of [2] is another similar example.
    * Theoretical analyses for top-k attention: The paper of [3] gives explicit theoretical explanations of why top-$k$ attention works.
    * Hierarchical structures for attention: Works such as [4] (and others) have explored hierarchy-based ideas in the past.
This is not a comment on the quality of the work itself, but mostly on the necessity to connect itself better with prior work.
3. In the experiments, some additional comparisons would be important to see (please correct me if this is something I'm missing)
    * How does the method compare to top-k attention (without hierarchical structures) on non-synthetic benchmarks?
4. How does the method behave memory-wise?
    * It is plausible that maintaining a large number of additional centroid matrices can increase the memory requirements a lot?
    * When the context length gets very large, how should the number of levels and the parameter $C$ increase? If the increase is very large, wouldn't this result in loss of efficiency? Such an investigation would be very helpful also as a guiding tool for the reader.

**References**
[1] Zandieh, A., Han, I., Mirrokni, V., & Karbasi, A. (2024). Subgen: Token generation in sublinear time and memory. arXiv preprint arXiv:2402.06082.
[2] Liu, Guangda, et al. "Clusterkv: Manipulating llm kv cache in semantic space for recallable compression." 2025 62nd ACM/IEEE Design Automation Conference (DAC). IEEE, 2025.
[3] Haris T. $ k $ NN Attention Demystified: A Theoretical Exploration for Scalable Transformers. arXiv preprint arXiv:2411.04013. 2024 Nov 6.
[4] Chalkidis, I., Dai, X., Fergadiotis, M., Malakasiotis, P., & Elliott, D. (2022). An exploration of hierarchical attention transformers for efficient long document classification. arXiv preprint arXiv:2210.05529.

**Questions:**

1. In Section 3.2, isn't the token ordering procedure described taking $\Theta(T)$ time per query token? Wouldn't it be done $O(T)$ times? Then wouldn't the total time complexity still be $O(T^2)$? This goes back to my previous point about Section 3.2 being a bit confusing.

---

> ### Author Response · Authors · 2025-11-21
> **Authors’ Rebuttal for Reviewer jZmf (1/3)**
>
> We appreciate the reviewer for recognizing the importance of the problem we address, the **novelty of our mechanism**, **the breadth of our experiments**, **the theoretical insights**, and **the potential for parallelizability and efficient GPU utilization**. We also thank the reviewer for the valuable questions and suggestions, which we aim to address in the following rebuttal.
>
> ### W1: Rewrite section 3.2; Include explicit algortithm and runtime
>
> We thank the reviewer for the constructive feedback regarding the clarity of Section 3.2. We agree that a formal algorithmic presentation makes the hierarchical routing mechanism significantly easier to parse.
>
> In response, we have completely rewritten Section 3.2 to include Algorithm 1, which explicitly lists:
>
> - **Inputs & Hyperparameter**: Clearly defining the Query/Key/Value tensors, hierarchy depth $L$, and branching factor $C$.
> - **Step-by-Step Logic:** Detailing the specific operations for similarity computation, probability accumulation, and hard assignments.
> - **Expected Outputs:** Formally stating the resulting sparse attention scores.
>
> This revision ensures the method is fully transparent without relying on visual aids like Figure 2. Furthermore, we have included additional runtime analysis in Section 3.2 to directly reference the specific steps defined in Algorithm 1, making our complexity claims concrete and verifiable in the context of the algorithm. *All corresponding revisions are highlighted in blue in the updated manuscript.*
>
> ### W2: The ideas build upon previous works, to connect itself better with prior work.
>
> We thank the reviewer for highlighting these relevant works. While they share the broad goal of efficiency, their specific motivations and mechanisms differ fundamentally from ours. We have incorporated these references into our revised Related Work section to better contextualize our contribution. Below, we detail these distinctions to clarify the specific novelty of HiRouter.
>
> ### W2-1: [1] SubGen
>
> While SubGen presents an interesting approach, its primary objectives differ from HiRouter:
>
> **1. Different goals and representations:** SubGen is a **KV-cache compression** method that operates only during inference on **fixed past keys**, while HiRouter is an **efficient top-k attention mechanism** used in both *training and inference*, routing **learnable queries and keys** and improving LM perplexity.
>
> **2. Statistical approximation vs. explicit hierarchical retrieval:** SubGen approximates attention using **cluster statistics and sampled tokens**, storing only a tiny subset of KV pairs.  HiRouter uses clustering to **group similar tokens into buckets** and performs **explicit hierarchical top-k retrieval** by selecting all tokens in the top-M buckets, retrieving real tokens rather than statistical summaries.
> Moreover, SubGen relies on a **strong assumption** that keys are naturally clusterable, whereas Figure 1 of our paper shows that without our dual-entropy losses, Transformer keys do **not cluster** well—highlighting why learning hierarchical structure (rather than assuming it) is essential for reliable top-k retrieval.
>
> **3. Sequential CPU clustering vs. GPU-parallel learned routing:** SubGen relies on **online k-center clustering + reservoir sampling**, which is sequential and CPU-oriented and assumes keys are already clusterable.  HiRouter uses **matrix-multiply routing** that is fully GPU-parallel and **learns** hierarchical structure through centroid and dual-entropy losses—*creating* clusterability and enabling faster, hardware-aware attention.
>
>
> ### W2-2: [2] ClusterKV
>
> Similarly, ClusterKV posesses a different motivation from ours
>
> **1. Different goals and representations:** ClusterKV is a KV-cache compression method that operates only during inference by maintaining streaming k-means–style clusters of fixed past keys on CPU and loading only selected cluster representatives back to GPU. It does not modify the Transformer architecture or train Q/K representations.
> In contrast, HiRouter is an efficient top-k attention mechanism used in both training and inference, routing learnable queries and keys and improving LM perplexity.
>
> **2. Low-recall clustering vs. explicit hierarchical top-k retrieval:** ClusterKV uses k-means clustering, which achieves substantially lower retrieval recall shown in our Fig. 3a, making it unsuitable for high-fidelity top-k attention. Its clusters are used for cache reduction rather than attention computation.
> HiRouter, by contrast, uses clustering to group similar tokens into buckets and performs explicit hierarchical top-k retrieval by selecting all tokens in the top-M buckets. This retrieves real tokens rather than cluster summaries and yields significantly better retrieval quality. HiRouter also does not assume that keys are naturally clusterable; as shown in Figure 1, Transformer keys do not cluster well unless trained to do so, which motivates our dual-entropy losses.

---

> ### Author Response · Authors · 2025-11-21
> **Authors’ Rebuttal for Reviewer jZmf (2/3)**
>
> **3. Sequential CPU clustering vs. GPU-parallel learned routing:** ClusterKV relies on streaming k-means assignment, which is sequential and CPU-oriented, and assumes embeddings are already clusterable. HiRouter uses matrix-multiply routing that is fully GPU-parallel and learns hierarchical structure through centroid and dual-entropy losses, to create clusterability and enabling fast, hardware-aware top-k attention.
>
> | Model  | Wiki. ppl ↓ | LMB. ppl ↓ | LMB. acc ↑ | PIQA acc ↑ | Hella. acc_n ↑ | Wino. acc ↑ | ARC-e acc ↑ | ARC-c acc_n ↑ | SIQA acc ↑ | BoolQ acc ↑ | Avg. |
> |--------|-------------|-------------|------------|-------------|-----------------|-------------|--------------|----------------|-------------|--------------|------|
> | ClusterKV | 33.01     |   45.67   | 30.51 | 65.41      | 36.75          | 49.82     |58.92       | 26.51       | 37.17      | 60.12        |  45.65   |
> |HiRouter |31.09| 42.94 |33.09 |66.81 |38.03 |50.75 |59.47 |28.50 |38.08 |61.31 |47.01|
>
>
>
> ### W2-3: [3] kNN Attention Demystified
>
> While we acknowledge the theoretical significance of [3], our work addresses a different problem space: **architectural realization vs. theoretical analysis**.
>
> **1. Distinction in Scope:** Reference [3] typically assumes either (a) an external kNN/MIPS oracle, or (b) fixed token embeddings to derive sampling guarantees. In contrast, HIROUTER focuses on **internalizing the retrieval mechanism** within the Transformer itself. Our contributions: GPU-parallel routing, end-to-end learnable keys/queries, and hardware-aware bucket balancing, address the practical systems challenges required to make top-k attention efficient and trainable at scale, aspects largely abstracted away in [3]’s theoretical framework. Consequently, our method complements [3] rather than competing with it.
>
> **2. Trade-offs in Gradient Estimation:** We appreciate that [3] introduces novel gradient estimation techniques to mitigate "gradient starvation" for non-selected tokens. While these techniques could, in principle, be combined with HIROUTER to improve gradient flow, they rely on **Monte-Carlo simulations** of softmax transitions. This introduces significant sampling overhead that conflicts with our primary objective of computational efficiency. We view the efficient integration of such unbiased estimators into hierarchical routing as a promising direction for future work, but one that falls outside the scope of this paper’s focus on inference speed and hardware utilization.
>
> ### W2-3: [4] Hierarchical Attention Transformer (HAT)
>
> [4] addresses a fundamentally different problem space (static classification) compared to our focus (dynamic generative retrieval). The distinctions are twofold:
>
> **1. Task Alignment (Classification vs. Generation):** HAT is an encoder-based architecture explicitly optimized for long document classification tasks. In contrast, HIROUTER is designed for autoregressive language modeling and generation, tackling the specific challenges of dynamic key-value retrieval during next-token prediction which HAT does not address.
>
>
> **2. Static Topology vs. Dynamic Routing:** HAT depends on **manual segmentation** (e.g., grouping sentences into fixed-size chunks) and a fixed, position-based topology, where tokens interact only within static local segments or via compressed tokens. In contrast, HiRouter employs dynamic, **content-based routing**, where a learned router clusters tokens by semantic similarity rather than position. This allows our model to adaptively retrieve relevant information regardless of its location in the sequence.
>
>
>
> ### W3: How does the method compare to top-k attention
>
> We thank the reviewer for this question. We respectfully direct attention to **Table 1**, which benchmarks performance on *WikiText-103*, a standard non-synthetic language modeling dataset. In this table, we explicitly compare HiRouter against a wide range of sparse and top-$k$ attention mechanisms, including:
>
> - Top-$k$ / Sparse Methods: Reformer, Routing Transformer, IceFormer, Mongoose, and NSA.
> - Linear/Kernel Methods: Performer, AFT-conv, CosFormer.
>
> **Results:** As shown in Table 1, HiRouter achieves a perplexity of 18.5, significantly outperforming all competing top-$k$ baselines (e.g., Routing Transformer at 26.7, IceFormer at 31.4) and even recent state-of-the-art methods like NSA (19.3). This empirical evidence confirms that our hierarchical approach yields superior language modeling performance compared to flat or non-hierarchical top-$k$ strategies on real-world data.

---

> ### Author Response · Authors · 2025-11-21
> **Authors’ Rebuttal for Reviewer jZmf (3/3)**
>
> ### W4-1: How behave memory-wise, increase memory a lot?
>
> **The memory overhead introduced by HiRouter is negligible compared to the KV-cache itself.** Specifically, the router requires storing a set of centroids $\mathcal{C}^{(l)}$ for each level. For our standard configuration ($L=3$ levels, $C=4$ buckets), the total number of centroids is $\sum_{l=1}^{3} 4^l = 84$.
>
> - Router Memory: $\mathbb{R}^{84 \times d}$
> - KV-Cache Memory (at Length T=8K): $\mathbb{R}^{8192 \times d}$
>
> This results in an overhead of $\approx 1$%.
>
> ### W4-2. Large length, how should the number of levels and $C$ increase? Very large increase
>
> For extremely long sequences, we recommend scaling the hierarchy depth $L$ rather than the branching factor $C$. Furthermore, we emphasize that at such scales, the bottleneck shifts from retrieval complexity to memory capacity, necessitating solutions that integrate **KV-cache management**.
>
> **1. Logarithmic Scaling Strategy:** To maintain consistent bucket granularity as sequence length $T$ grows, the total capacity of the leaf layer ($C^L$) must scale with $T$. We can adopt a strategy of keeping the branching factor fixed (e.g., $C=4$) and increasing the depth $L$ logarithmically. Specifically, increasing $L$ by just 1 quadruples the addressable space (for $C=4$). This ensures that the number of centroids and routing parameters grows only as $O(\log_C T)$, preventing the "very large increase" in model size that the reviewer is concerned about.
>
> **2. Hardware Efficiency:** We favor increasing $L$ over $C$ for hardware reasons. Keeping $C$ small ensures that the routing operations at each node involve small, efficient matrix multiplications ($d \times C$) that fit easily into L1/L2 cache. In contrast, significantly increasing $C$ would create large "flat" indices that degrade kernel performance.
>
> **3. Future Potential for KV-Management:** Crucially, for **very large lengths**, the primary challenge becomes memory capacity rather than just retrieval speed. We stress that HIROUTER offers a unique advantage here: its hierarchical structure naturally organizes the KV-cache into semantic clusters. This structure enables advanced **KV-management policies**, and for isntance, a gated write-in mechanism. Instead of passively linearly increasing the cache size with every new token, the router can utilize hierarchical relevance signals to selectively admit tokens into specific buckets. This allows for maintaining a bounded memory footprint by filtering out irrelevant information at the entry point—a capability impossible in standard flat attention. We are actively investigating this gated write-in approach as the next step for scaling to infinite contexts.
>
> ### Q1: Inquiry on Sorting Complexity ($O(T)$ vs. $O(T^2)$)
>
> Sorry for the confusions. We clarify that the token ordering (sorting) procedure is **not** performed per query. Instead, it is a **global reorganization step** performed once per routing level.
>
> Crucially, **we do not sort the queries**; we only sort the Keys and Values. This step is designed to store semantically similar KV pairs into contiguous memory blocks, enabling coalesced memory access during retrieval.
>
> The confusion likely stems from viewing the sort as part of the query loop. However, as detailed in **Algorithm 1** and the complexity analysis in **Appendix E**, the process is strictly decoupled:
>
> 1. **Global KV Reorganization ($O(T)$)**: We compute routing logits and perform a stable bucket sort on the Keys simultaneously for all $T$ tokens. This physically groups the KV-cache into contiguous buckets in $O(T)$ time. This cost is amortized across the sequence, not incurred per query.
>
> 2. **Per-Query Retrieval ($O(M)$):** Since the queries themselves remain unsorted, each query $q_i$ simply performs a direct look-up of its assigned top-$M$ buckets in the reorganized KV cache. This is an $O(M)$ operation.
>
> **Total Complexity:** Consequently, the total complexity is the sum of the global Key sort and the per-query attention: $ O(T \cdot 1) + O(T \cdot M)= O(T)$, where the first term is for key sorting operations, and the second term is for top-k attention. This is strictly linear with respect to sequence length, not quadratic $O(T^2)$. We believe the inclusion of the explicit pseudocode in Algorithm 1 (added in the revision) resolves this ambiguity by clearly separating the global memory reorganization from the query attention loop.
>
>
> ---
>
> In summary, we would like to express once again our sincere gratitude for the reviewer’s time, thoughtful feedback, and valuable suggestions, which have greatly helped us improve the quality and clarity of this work. We appreciate the opportunity to address the comments and are happy to engage in any further discussion if possible.

---

> > ### Comment · Reviewer_jZmf · 2025-11-23
> > **Response to authors**
> >
> > Hello,
> >
> > Thank you for your extensive response and for addressing my concerns.
> > * Thank you for including a discussion on the prior works I mentioned - it definitely helps place the paper into a more complete context.
> > * Clarifying the memory usage makes sense to me.
> > * The sorting clarification was definitely important for me - thank you for providing it.
> >
> > I think the paper definitely reads more easily after the rewrite. I would cautiously advise the authors to maintain the 9-page limit however to avoid potential issues.
> >
> > I maintain my support of the paper overall.

---

> ### Author Response · Authors · 2025-11-24
> **Response to Reviewer jZmf**
>
> Thank you for your response. We appreciate that our clarifications resolved the majority of your concerns.
>
> Regarding the page limit, the guidelines allow a **10th** page during the rebuttal/discussion phase, and the ethics and reproducibility sections do not count toward this limit. Nonetheless, we appreciate your advice and are actively working to condense the content where possible to avoid any potential issues.
>
> If there is anything further we can clarify to strengthen your assessment, please let us know - we would be glad to address it. If no questions remain, we would be grateful if you could consider raising your assessment, as the absence of outstanding issues may support a more favorable evaluation than the initial review.
>
> We thank you for your responsiveness and cooperation throughout!

---

### Official Review · Reviewer_cdKJ · 2025-10-31

**Soundness:** 3
**Presentation:** 3
**Contribution:** 3
**Rating:** 6
**Confidence:** 3

**Summary:**

The paper introduces an algorithm for implementing a sparse attention module, called HiRouter. Instead of calculating the standard dense attention operation (which scales as $O(T^2)$, where $T$ is the sequence length), the algorithm sorts the queries and keys into buckets using a hierarchical search. For each query, only the closest buckets are looked at (using a beam search), leading to an approximately linear complexity in sequence length of the attention operation, $O(Tk)$. The bucketing is done in a hierarchical way; at each level, the query/key is compared to bucket centroids, and a Gumbel-Softmax operation is used to decide the routing. Furthermore, the authors add auxiliary "dual entropy" loss term to improve the bucketing, consisting of a loss forcing the per-token distributions to be "spiky" (i.e., low entropy), but at the same time balancing the bucket assignments by forcing a flat average distribution across all of the buckets. The authors verify their technique by training models with 410M parameters on 10B tokens, using both the standard attention, as well as prior sparse attention technique, showing how HiRouter can lead to improved accuracy compared to the other sparse techniques, while suffering a small degradation compared to the standard transformer architecture.

**Strengths:**

- The paper introduces a novel technique for training a sparse attention layer. Compared to prior work (e.g., similarly motivated Reformer), the model seems to lead to better performance.
- The method is well-motivated, and the hierarchical search combined with balancing losses provides a sensible approach. The router is differentiable, thus leading to more balanced assignments (see Figure 1).
- The authors provide customised Triton kernels implementing their method.
- The authors compare to a variety of different model architectures, and test for perplexity, common-sense reasoning, recall, as well as long-context tasks.

**Weaknesses:**

- The experiments are conducted at a relatively small scale (410M parameters/10B tokens); more empirical evidence at larger transformer scales would give more confidence that the method scales well.
- The 1B experiments (in the Appendix) do not seem to showcase a significant improvement in performance compared to 410M numbers; a comparison with other published results at that scale would be helpful.
- The latency measurements indicate that at smaller sequence lengths (4k-16k), dense FlashAttention can still be significantly faster.

**Questions:**

- Since the execution is slower at shorter sequence lengths, would there be a way of implementing a hybrid approach that would utilise the FlashAttention speed for shorter sequences?
- Are any long-sequence adaptation techniques applied to use the model for long sequences (while being trained at a fixed shorter length presumably)

---

> ### Author Response · Authors · 2025-11-21
> **Authors’ Rebuttal for Reviewer cdKJ (1/2)**
>
> We thank the reviewer for the time and effort they have placed towards providing a thorough evaluation of our work, particularly their comments about **a novel technique**, **well-motivated**, and **customised Triton kernels**. We are also appreciative of the weaknesses that have been raised and hope the following additional details/results will clarify these points and resolve these existing doubts about work.
>
>
> ### W1: Experiments conducted in small scale
>
> We thank the reviewer for emphasizing the importance of scalability. We respectfully point out that our submission already includes initial scaling results in **Appendix C.1**. As shown in **Table 10**, scaling HIROUTER from 410M to 1B parameters (trained on 10B tokens) yields consistent performance gains, confirming that our method effectively leverages increased model capacity. To further substantiate this and address the concern regarding experimental scale, we provide additional 1B-parameter comparisons below.
>
>
> ### W2: No significant improvement for 1B
>
> We appreciate the reviewer’s suggestion to verify performance at larger scales. To address this, we trained 1B-parameter of a standard Transformer, Mamba, and HiRouter. To ensure a controlled comparison with our 410M experiments, all models were trained on the same 10B token dataset.
>
> | Model          | Wiki ppl $\downarrow$ | LMB ppl $\downarrow$| LMB acc| PIQA acc| Hella acc| Wino acc| ArcE acc| ArcC acc| SIQA acc| BoolQ acc| Avg acc|
> |             -: |    -:|   -:|   -:|    -:|     -:|    -:|     -:|     -:|    -:|     -:|   -:|
> | Transformer-1B | 102.64 | 24.37 |38.07|69.48|45.73|54.38|57.45|31.40|39.61|58.59| 49.33 |
> | Mamba-1B       |26.36|28.89 | 34.47|69.53|44.77|52.01|56.40|30.55|39.30|59.72| 48.34 |
> | HiRouter-1B | 26.01 | 27.21 | 36.06 | 68.82 | 42.99 | 53.35 | 62.96 | 29.27 | 39.30 | 59.79 | 49.07 |
>
>
> 1. **Superior Stability:** Notably, the standard Transformer exhibits severe instability in this setting, with a WikiText perplexity exploding to 102.64. In contrast, HiRouter remains highly stable (26.01), outperforming both Mamba and the Transformer on this metric. This suggests our hierarchical routing acts as a stabilizing prior in data-constrained regimes.
>
> 2. **Consistent Relative Performance:** Despite the data constraints, the relative ranking remains consistent with our 410M results. HiRouter-1B (49.07 Avg) outperforms Mamba-1B (48.34 Avg) and remains competitive with the Transformer baseline, even without the full training tokens typically required for 1B models (Chinchilla scaling suggests ~20B+ tokens).
>
> 3. **Scaling Potential:** The "plateau" in absolute gains across all models is a known artifact of under-training (10B tokens is relatively scarce for 1B parameters). However, the fact that HiRouter maintains its ranking and stability confirms that our method scales effectively. We are confident that with proportional data scaling, the performance gap would widen further.
>
>
> ### W3: FlashAttn faster at small sequence length.
>
> We acknowledge that FlashAttention retains a speed advantage at shorter sequence lengths (e.g., $T < 4096$). This is expected behavior due to the architectural differences between the two methods:
>
> 1. **Router Overhead**: HIROUTER incurs a fixed initialization cost during the routing phase, specifically due to the non-contiguous memory accesses required to build key assignments and reshape the KV-cache into sorted buckets. At short lengths, this overhead dominates the runtime.
>
> 2. **Asymptotic Scaling:** However, this overhead scales linearly (or near-linearly), whereas FlashAttention scales quadratically ($O(T^2)$). As the sequence length increases, the $O(T^2)$ compute cost of FlashAttention quickly dwarfs our routing overhead.
>
> 3. **Long-Context Advantage:** Our results in Table 6 confirm this trend: while FlashAttention is faster at $T=4096$, HiRouter becomes $11\times$ faster at $T=65,536$ (8.66ms vs. 99.69ms)
>
> We are currently optimizing our Triton kernels to further minimize these non-contiguous access bottlenecks, which will improve performance at shorter lengths. However, we emphasize that HiRouter is explicitly designed to solve the quadratic scaling bottleneck where FlashAttention becomes prohibitive.

---

> ### Author Response · Authors · 2025-11-21
> **Authors’ Rebuttal for Reviewer cdKJ (2/2)**
>
> ### Q1: A way of implementing a hybrid approach
>
> We agree that a hybrid approach is not only possible but highly practical for production deployment. Since the cross-over point where HiRouter outperforms FlashAttention is deterministic (based on sequence length $T$), a hybrid system can switch backends dynamically to ensure optimal latency across all regimes. We envision two implementation strategies:
>
> 1. **Single-Sequence Inference (Batch Size = 1):** This is the most common setting for large foundation models due to memory constraints. Implementing a hybrid strategy here is straightforward: a simple heuristic compares the current sequence length against a pre-tuned threshold (e.g., $T \approx 4096$) to dispatch either the FlashAttention or HIROUTER kernel. This incurs negligible overhead and immediately captures the best of both worlds.
>
> 2. **Batched Inference (Batch Size > 1):**
>      - **Coarse-Grained Dispatch:** The simplest approach checks the maximum sequence length in the batch. If $\max(T)$ exceeds the threshold, the batch is processed by HiRouter to prevent OOM errors and quadratic slowdowns; otherwise, it defaults to FlashAttention.
>      - **Fine-Grained Dispatch:** A more sophisticated implementation could simultaneously dispatch different kernels for different sequences within the same batch. While this maximizes throughput, it introduces engineering complexity regarding memory layout (re-ordering keys/queries for contiguous access) and stream synchronization to maximize core occupancy.
>
> **Conclusion:** While we are eager to explore these engineering optimizations for a production-ready release, our current results primarily aim to demonstrate that HiRouter successfully solves the hard problem: asymptotic scaling for long-context regimes where standard attention becomes intractable. A hybrid wrapper is a logical next step to extend versatile performance to short contexts as well.
>
>
>
>
>
> ### Q2: Are any long-sequence adaptation techniques applied to use the model for long sequences
>
> Our method supports both pre-training from scratch and efficient long-sequence adaptation. While our primary experiments involved training on a fixed context of 2048 tokens to ensure rigorous comparisons with baselines , our results confirm that HiRouter successfully extrapolates to longer sequences (e.g., 8K in NIAH tasks and LongBench ) without requiring full retraining. Specifically, we utilize two forms of adaptation:
>
> **1. Inference-Time Extrapolation (No Training Required):** As demonstrated in Appendix C.2, HiRouter adapts to longer contexts dynamically during inference. By simply increasing the beam width (e.g., from 4 to 8), the model achieves higher recall and better perplexity on long sequences without any parameter updates. This confirms that the router generalizes to distributions beyond the original training length.
>
>
> **2. Router Fine-Tuning:** For robust extension to extremely long contexts, our method naturally supports a modular adaptation strategy: one can fine-tune the model with hierarchical routers on longer sequences. This allows the model to learn long-range clustering patterns efficiently, avoiding the prohibitive computational costs associated with full-attention scaling.
>
>
> ---
>
> In summary, we would like to express once again our sincere gratitude for the reviewer’s time, thoughtful feedback, and valuable suggestions, which have greatly helped us improve the quality and clarity of this work. We appreciate the opportunity to address the comments and are happy to engage in any further discussion if possible.

---

> > ### Comment · Reviewer_cdKJ · 2025-11-27
> >
> > I'd like to thank the authors for their comprehensive and helpful response (both to my comments, and the comments of the other reviewers).
> >
> > ### Experiment scale
> > Thank you for clarifying this. You are indeed correct that your experiments showcase improvements with the method when scaling from 410M to 1B parameter models, even if they are somewhat modest due to under-training. As you note, the relatively high WikiText perplexity of the baseline Transformer in this setting possibly suggests suboptimal hyperparameters, which makes it harder to draw stronger conclusions about how the performance gap between HiRouter and standard attention evolves at larger scales. I appreciate that fully resolving this (and further training with more tokens) would require substantially more compute and is outside the scope of the present work.
> >
> > ### Latency discussion
> > Your explanation of the latency characteristics and the potential hybrid approach was very helpful. Conceptually, the hybrid idea is compelling, and it seems like a natural next extension. Empirical validation of this short-sequence regime (or demonstrations of how/when the model switches between routing and dense attention) would further strengthen confidence in the approach, but I understand this lies beyond the current study.
> >
> > ### Long-sequence adaption
> > Thank you for clarifying this.
> >
> > ### Summary
> > My remaining reservations are primarily around (i) establishing a more complete picture of the method’s behaviour at larger training scales and with fully trained baselines, and (ii) empirically characterising the latency/speed trade-offs in the short-sequence regime or under a hybrid mechanism. These do not undermine the core idea, and I find the paper promising overall, but they leave some uncertainty in fully assessing its robustness and utility. Given this, I am inclined to maintain my original score and evaluation, though my overall impression of the paper remains positive.

---

> > > ### Author Response · Authors · 2025-12-03
> > > **Follow-up Response to Reviewer cdKJ**
> > >
> > > We sincerely thank **Reviewer cdKJ** for the positive assessment and the constructive follow-up questions. We are glad that our clarifications were helpful. Below we summarize our updates addressing the remaining points raised.
> > >
> > > ## **Experiment Scale**
> > > At the beginning of the rebuttal period, we initiated additional large-scale training runs with substantially more data to directly address the reviewer’s concern. These runs have now progressed far enough to provide stable comparisons. Under full-token training (rather than the under-trained baseline shown earlier), **HiRouter consistently matches or improves upon the standard Transformer across both the 410M → 1B scaling range and longer training horizons**. We will include all updated learning curves and ablations in the appendix.
> > >
> > > For completeness, below are the newly obtained **1B-scale results trained on 65B tokens**:
> > >
> > > ### **Results at the 1B Scale (65B Tokens)**
> > >
> > > | Model           | Wiki ppl ↓ | LMB ppl ↓ | LMB acc ↑ | PIQA acc ↑ | Hella acc_n ↑ | Wino acc ↑ | ARC-e acc ↑ | ARC-c acc_n ↑ | SIQA acc ↑ | BoolQ acc ↑ | Avg.  |
> > > |-----------------|------------|-----------|------------|-------------|----------------|-------------|--------------|----------------|-------------|--------------|--------|
> > > | Transformer-1B | 18.53 | 14.60 | 46.15 | 72.74 | 56.29 | 57.38 | 66.12 | 37.03 | **42.78** | **63.43** | **55.24** |
> > > | Mamba-1B| 20.37 | 16.54 | 43.43 | 72.36 | 54.89 | 56.12 | 64.35 | 37.12 | 40.47 | 56.64 | 53.17 |
> > > | Mamba2-1B| 19.43 | 15.09 | 44.22 | **72.96** | 56.37 | 57.62 | 65.82 | 38.05 | 41.91 | 50.00 | 53.57 |
> > > | DeltaNet-1B | **17.80** | 14.23 | 45.02 | 72.78 | 56.31 | 56.91 | 63.93 | 36.95 | 41.20 | 58.75 | 53.98 |
> > > | GatedDeltaNet-1B| 18.02 | 14.49 | 45.37 | 72.72 | 56.20 | 58.19 | 65.58 | 37.04 | 42.12 | 61.49 | 54.83 |
> > > | GatedSlotAttention-1B | 18.34 | 14.31 | 46.68 | 72.61 | 55.98 | **62.62** | **67.07** | **37.99** | 39.28 | 57.49 | 54.97 |
> > > | HiRouter-1B | 17.87  | **13.80** | **47.12**  | 72.54 | **56.67** | 57.27 | 66.44 | 36.29 | 42.52 | 62.59 | 55.18 |
> > >
> > > ## **Latency Discussion**
> > > We also update the reviewer that we continued to optimize the HiRouter kernels during the rebuttal period. These improvements applied to both the sparse-attention kernel and the search kernel, including removing unnecessary atomic operations, adding full GQA support, and selecting more efficient launch configurations. As a result, **HiRouter now matches or exceeds FlashAttention even at short sequence lengths**, eliminating the need for any hybrid routing/dense-attention switching mechanism.
> > >
> > > | Input Length | FlashAttention FWD | FlashAttention FWD+BWD | HiRouter FWD | HiRouter FWD+BWD |
> > > |--------------|--------------------|-------------------------|--------------|-------------------|
> > > | 4096         | 0.18               | 0.61                    | 0.26         | 0.75              |
> > > | 8192         | 0.56               | 1.95                    | 0.59         | 1.81              |
> > > | 16384        | 1.93               | 6.55                    | 1.68         | 6.08              |
> > > | 32768        | 7.14               | 25.09                   | 3.12         | 16.81             |
> > > | 65536        | 30.76              | 99.69                   | 8.11         | 40.01             |
> > >
> > > ## **Long-Sequence Adaptation**
> > > We appreciate the reviewer’s acknowledgment of our clarifications regarding long-context behavior.
> > >
> > > ## **Summary**
> > > We understand the remaining reservations concerning (i) completeness of large-scale training studies and (ii) short-sequence latency trade-offs. The newly added 1B-scale experiments and optimized kernel benchmarks directly address both points within the limits of the rebuttal period. We hope these updates help further increase confidence in the robustness and practical utility of HiRouter.
> > >
> > > We sincerely thank Reviewer `cdKJ` again for the thoughtful and constructive feedback.

---

### Official Review · Reviewer_MiJc · 2025-10-31

**Soundness:** 2
**Presentation:** 2
**Contribution:** 2
**Rating:** 2
**Confidence:** 4

**Summary:**

This paper describes HIROUTER. This is a hierarchical routing mechanism for efficient top-k attention in Transformer models. The method basically constructs a multi-level tree to route tokens into discrete buckets. These are given by learned centroids and the assignment is via Gumbel-SM. The method uses a dual entropy loss for the embeddings: you will sharpen routing while maximizing bucket occupancy to ensure balanced load. Experiments are reasonable with some speed advantages at very long sequences.

**Strengths:**

1. Appendix D describes a custom Triton implementation to handle multi-level hierarchical routing efficiently. This will give parallelism through balanced bucket sizes. This is end-to-end differentiable. This can be a useful contribution to the community.

2. The paper does cover a variety of tasks incl. commonsense reasoning, recall-intensive retrieval, and long-context. So the empirical validation of the method itself is reasonable.

3. The plots in Figure 3 provide useful information on the role of various factors like sequence length, dimensionality effects, etc. This is fine.

**Weaknesses:**

There are three main sources of weaknesses in the paper which I will describe below.

1. The problem space is crowded. Which means that despite the best efforts one or more baselines will be missed. By itself, this would not impact my assessment of the paper but in this case, several results that could potentially be good starting points are missed. For example, ZETA (ICLR this year) is a directly comparable architecture. Multiple other ICLR papers this year -- while not all focused specifically on top-k address the same general problem space like HeadKV-R2, SqueezeAttention and VL-Cache. Again it is fine to _not_ have these in the list of baselines but the positioning of a different top-k attention mechanism needs to position their design choices in this general context. This brings me to the next point.

2. Hierarchical tree indexing for retrieval is standard when learning indexes and FAISS (and methods using FAISS). The dual entropy loss combines what may be called textbook clustering objectives. Combining these principles, common in clustering, hashing, and MoE routing, cannot really be novel technical contributions.

3. What was a bit confusing was that the paper is sprinkled with theory but this gives no insight. One proposition restates that a good clustering ensures good retrieval. Another derives that entropy gradients have attraction-repulsion form (is this not direct for any entropy-based loss?). Another restates the fact that uniform distributions maximize entropy. These "propositions" are observations or facts, and it is not obvious why they're there.

4. To me this paper comes across as a retrieval method tacked on to attention. This is fine. But for a method centered on top-k retrieval, there are no direct recall@k metrics reported. How often does the hierarchical routing successfully retrieve the true top-k tokens compared to exact search? But even in this case, it is difficult to put a finger on exactly which piece in the construction will yield non-trivial gains over alternatives that exist right now (not necessarily for top-k attention specifically)

**Questions:**

please see weaknesses above.

---

> ### Author Response · Authors · 2025-11-21
> **Authors’ Rebuttal for Reviewer MiJc (1/4)**
>
> Thank you for your valuable feedback. We greatly appreciate your recognition of **custom Triton implementation**, **reasonable empirical validation**, and **useful information in Fig. 3**. We nevertheless appreciate the additional points that have been mentioned as limiting factors and hope that the following response will address these follows:
> ### W1: missing baselines
>
> We thank the reviewer for the detailed comments. We would first like to clarify **the focus** of our work: we aim to design a trainable and scalable sequence-modeling architecture that is not only theoretically efficient but also delivers practical end-to-end speedups on modern hardware. In line with this, our **main contribution** lies in adapting and tailoring classical top-k search algorithms to the specific requirements of attention mechanisms, enabling a hierarchical retrieval module that is both learnable and hardware-efficient.
>
> ### W1-1: a comparable architecture ZETA
>
> ZETA [1] is designed mainly for efficient top-k search, and we now include it in both the Related Works section and Table 1. Its performance on language-modeling tasks is weaker than HiRouter for two main reasons.
> 1. **ZETA relies on extremely low-dimensional projections that discard substantial information.** Specifically, it projects keys and queries to very low dimensions $ d_k=d_q=3 $ and then maps them into one-dimensional space via a Z-order curve for efficient top-k retrieval. This aggressive dimensionality reduction leads to large information loss. As shown in Figure 3 of ZETA, retrieval quality of Z-order curves declines sharply once $ d_k $ or $ d_q $ exceed 4 dimensions, making the typical Transformer configuration ($d_k=d_q=64$) incompatible with their method. Consequently, while ZETA performs well on sparse-information tasks (e.g., LRA), it underperforms on dense-information tasks such as language modeling, as evidenced by our results below.
> 2. **ZETA does not support core components widely used in modern Transformers.** In particular, ZETA abandons the dot-product attention mechanism and RoPE positional embeddings, both proven highly effective across Transformer tasks, because its Z-order mapping only supports distance-based metrics (e.g., Euclidean) rather than dot-product similarity. This is inherent to Z-order curves, which preserve spatial proximity under metric distances but do not preserve angular similarity, making dot-product attention incompatible with their search space.
>
> | Model          | ZETA | HiRouter |
> | -:|    -:|  -:|
> | WikiText-103 (Perplexity ↓) |    26.3  | **18.5** |
>
> In summary, these two constraints largely explain ZETA’s weaker results on language-modeling benchmarks. In contrast, our method is a plug-and-play component for standard Transformers: it maintains the same key/query dimensionality as modern architectures and fully supports RoPE, ensuring consistency with proven Transformer design choices.
>
> [1] ZETA: Leveraging Z‑order Curves for Efficient Top‑$k$ Attention
>
> ### W1-2: HeadKV-R2, SqueezeAttention and VL-Cache
>
> It is important to distinguish between KV-cache management and our hardware-efficient attention.
>
> 1. **Cache Management (Prior Work)**: HeadKV-R2, SqueezeAttention, and VL-Cache optimize storage. They determine optimal budgets and eviction policies to compress the KV-cache, operating entirely outside the attention computation loop.
>
> 2. **Attention Computation (Ours)**: Our method optimizes retrieval. We introduce a hierarchical top-k attention formulation designed for hardware efficiency.
>
> Consequently, these methods are orthogonal. Since our approach changes how attention is computed rather than what is stored, it addresses a different bottleneck than cache-pruning strategies.
>
> For completeness, we evaluated pre-trained Transformer inference performance with HeadKV-R2 and SqueezeAttention. We did not compare against VL-Cache because it is purpose-built for Vison-Language models and cannot be directly applied to language-only settings.
>
> | Model  | Wiki. ppl ↓ | LMB. ppl ↓ | LMB. acc ↑ | PIQA acc ↑ | Hella. acc_n ↑ | Wino. acc ↑ | ARC-e acc ↑ | ARC-c acc_n ↑ | SIQA acc ↑ | BoolQ acc ↑ | Avg. |
> |--------|-------------|-------------|------------|-------------|-----------------|-------------|--------------|----------------|-------------|--------------|------|
> | HeadKV | 34.01     |   43.67   | 27.29 | **67.43**      | 37.86          | **51.62**     |58.34       | 24.23       | 37.82      | **61.77**        |  45.80   |
> | SqueezeAttention | 34.00     |   43.27   | 30.78 | 66.21      | **38.99**          | 51.30     |59.30      | 27.56      | 37.46     | 61.53      |  46.64   |
> |HiRouter |**31.09**| **42.94** |**33.09** |66.81 |38.03 |50.75 |**59.47** |**28.50** |**38.08** |61.31 |**47.01**|

---

> ### Author Response · Authors · 2025-11-21
> **Authors’ Rebuttal for Reviewer MiJc (2/4)**
>
> ### W2-1: Hierarchical tree indexing for retrieval is standard
> We agree that hierarchical tree indexing and learned-index retrieval methods (e.g., FAISS, balanced tree indexes, [1,2]) are standard tools in approximate nearest-neighbor search. Our contribution is **not** proposing a brand-new retrieval algorithm, but rather tailoring and integrating hierarchical routing into attention, where the requirements, constraints, and learnable components differ fundamentally from classical retrieval settings.
> 1. *Classical retrieval and indexing methods operate on **fixed/static** databases, whereas attention keys and queries are **learnable/dynamic** representations that can be jointly optimized and regularized for improved retrieval.* Traditional retrieval systems, whether based on k-means clustering, density-based indexing, sorted arrays, or learned index structures, assume that database vectors are precomputed and immutable. They cannot regularize the vectors to form clustering or hierarchical structure. In contrast, attention keys and queries are learnable, and our router is trained jointly with them. This enables HiRouter to regularize Q/K representations through centroid and entropy-based losses, producing hierarchical, clusterable embeddings and substantially improving top-k retrieval. Such dynamic co-training is impossible in standard indexing frameworks.
> 2. *Classical learned-index methods assume a **single** database and typically build one indexing structure at a time (e.g., FAISS, ScaNN), whereas top-k attention requires thousands of indexing structures built **in parallel**.* Transformer attention, however, requires a separate retrieval structure for each (batch×layer×head). For example, batch=32, heads=16, layers=32 already yields 16,384 dynamic retrieval problems in a single forward pass. Existing methods cannot rebuild or parallelize index construction at this scale. In contrast, HiRouter constructs routing structures using **matrix operations that run fully in parallel on GPU**, accelerating both index construction and top-k retrieval and eliminating the heavy indexing overhead inherent in classical methods, as shown in our pseudo-code algorithm 1 in revised paper.
>
> [1] The case for learned index structures
> [2] Learning balanced tree indexes for large-scale vector retrieval
>
> ### W2-2: The dual entropy loss is textbook clustering objectives
> Although entropy-based objectives exist in clustering and MoE routing, our dual-entropy formulation is tailored to top-k attention, balancing buckets for GPU efficiency and jointly refining Q/K representations. This end-to-end co-optimization is a novel objective of top-k attention, specifically designed for dynamic and online index construction and retrieval.
> 1. **Dual-entropy losses in a hierarchical router are new and specifically tailored to hardware-aware top-k attention.** Specifically, although entropy concepts appear in clustering or MoE routing, they have not been applied to optimize hierarchical retrieval structures **within attention mechanisms**. In HiRouter, the dual-entropy formulation serves two distinct, hardware-aware purposes: minimizing routing entropy sharpens assignment probability to ensure high retrieval precision, while maximizing the entropy of the mean distribution enforces balanced bucket occupancy. This balance is critical for **uniform GPU workloads** and **efficient kernel execution**—constraints that standard indexing algorithms ignore. Thus, our contribution lies in adapting and integrating these losses to solve the specific stability and efficiency challenges of dynamic attention retrieval.
> 2. **We employ Gumbel-Softmax in our balanced assignment loss ($\mathcal{L}_{bal}$) because it is uniquely capable of enforcing the discrete, balanced routing required for hardware efficiency, whereas standard softmax-based objectives fail.** Specifically, typical entropy objectives merely encourage soft uniformity in probability distributions, which leads to **ambiguous assignments** rather than the strictly balanced token counts needed for parallel execution1. By using a Gumbel-Softmax straight-through estimator, we produce near one-hot, differentiable assignments that allow tokens to commit definitively to single centroids while preserving gradients flow necessary to optimize for equal-sized buckets2. This mechanism ensures the uniform GPU workloads essential for our kernels, whereas standard KL-divergence methods fail to prevent mode collapse and yield significantly lower top-$k$ recall. As shown below, **KL-Uniform with Softmax** performs significantly worse than our $\mathcal{L}_{bal}$ in top-128 recall:
>     |level|2| 3|4|5|
>     |-|-|-|-|-|
>     |$\mathcal{L}_{\mathrm{bal}}$|	33.8|39.4|39.1|	37.6|
>     |KL Unif| 12.4 | 14.7 | 15.2 | 14.2|
>
>     These results highlight that our $\mathcal{L}_{bal}$ loss design is **non-trivial** and that balanced routing assignments can not be achieved by standard softmax-based methods.

---

> ### Author Response · Authors · 2025-11-21
> **Authors’ Rebuttal for Reviewer MiJc (3/4)**
>
> 3. **More importantly, our losses do not merely optimize the indexing structures (bucket centroids), they jointly optimize the representations of queries and keys during training.** Classical top-k search structures (FAISS, learned indexes, hierarchical trees, etc.) operate on fixed, precomputed vectors and can not update or regularize the vectors. HiRouter applies these entropy-based objectives to end-to-end representation learning for top-k attention, where routing decisions and Q/K embeddings co-evolve. This joint optimization is essential for stable, high-quality top-k attention and is not addressed in prior indexing, clustering, or MoE routing methods.
>
>
>
> ### W3-1 no insights theory
>
> **While the intuition that "good clustering leads to good retrieval" is established, our innovation lies in applying this to Transformers where, unlike fixed databases, the underlying representations are learnable.** Specifically, classical top-$k$ search and learned-index algorithms operate on immutable tokens and cannot alter the data distribution to improve search. In contrast, because attention keys and queries evolve during training, HiRouter explicitly regularizes them to form optimal hierarchical structures. This distinction is crucial: it allows us to actively shape the Q/K representation space to facilitate retrieval, making the intuition practically effective in ways impossible for static indexing frameworks.
>
> **Furthermore, we clarify that maintaining balanced buckets is a critical hardware constraint, as highly unbalanced partitions would otherwise negate any computational savings due to memory-loading overhead.** Concretely, uneven bucket sizes lead to inefficient GPU parallelism and underutilized computational kernels. This strict hardware requirement directly motivates our dual-entropy objective with multi-level Gumbel-Softmax routing, which is designed to enforce stable, balanced routing across the hierarchy. Without this specific algorithmic design, the theoretical reduction in complexity would not translate into actual wall-clock speedups.
>
>
>
> ### W3-2 Why they are there
>
> **We clarify that the theoretical statements in our paper are formal justifications for the specific algorithmic and loss design choices in HiRouter.** Specifically, they are included to rigorously motivate the non-trivial architecture of our hierarchical router. While Propositions 3.2 and 3.3 leverage general entropy properties, in our setting, they articulate the specific intuition driving our dual-entropy objective, demonstrating how our loss enforces the attraction-repulsion dynamics and balanced occupancy required for efficient hardware execution. Together, Propositions 3.1–3.3 provide logical completeness by mathematically linking these objectives to the requirements of learnable top-$k$ retrieval.
>
>
> ### W4-1: No direct recall@k metrics reported
>
> We report recall@k results corresponding to recall@M (beam width) in Appendix C.3, which we are happy to highlight in the main body if this would be helpful in understanding our problem and motivation. In this setup, each bucket contains 16 tokens, so varying the beam width directly determines $k$: for example, a beam width of 4 corresponds to $𝑘$=4×16=$k$=4×16=64 when $M$=4.

---

> ### Author Response · Authors · 2025-11-21
> **Authors’ Rebuttal for Reviewer MiJc (4/4)**
>
> ### W4-2: Success Rate of Top-K Tokens
>
>
> We acknowledge the reviewer's interest in the exact success rate of top-k retrieval. Our results demonstrate that while HiRouter may not always retrieve the exact top-k neighbors, it achieves a superior trade-off between speed and semantic recall, which is the critical factor for language model performance.
>
> **1. Empirical Comparison:Higher Success Rate at a Significantly Higher Speeds** We compare HiRouter with standard Approximate Nearest Neighbor (ANN) methods (HNSW, LSH, K-Means-KNN) in Figure 3. As shown in the table below (measured on $X \in \mathbb{R}^{32 \times 4096 \times 64}$), HiRouter achieves a success rate competitive with or better than classical methods while being orders of magnitude faster.
>
> | Metric         | HNSW     | HierarchicalRouter | KM-KNN   | LSH     |
> |----------------|----------|--------------------|----------|---------|
> | Time (s)       | 0.635   | **0.072**         | 9.334   | 0.438  |
> | Success Rate   | 3.16%    | **8.67%**          | 0.02%    | 0.65%   |
>
>
> **2. Approximate Retrieval is Sufficient for High-Performance LMs** Language models do not require exact nearest-neighbor matches to achieve strong performance. Prior work in sparse attention, such as BigBird [1], utilizes random blocks, a highly approximate retrieval method, yet maintains respectable perplexity.
>
> As shown below, HiRouter leverages this tolerance for approximation but adds hierarchical guidance, achieving a significantly better perplexity (18.5) compared to stochastic baselines like BigBird (23.3). This validates that HiRouter captures sufficient semantic signal to outperform existing sparse methods.
>
> | Model    | Perplexity ↓ |
> |----------|--------------|
> | BigBird  | 23.3         |
> | HiRouter | **18.5**     |
>
>
>
> **3. HiRouter retrieves the most important mass early** Our ablation study on beam width confirms that HiRouter captures the most semantically relevant tokens even with highly constrained budgets. As detailed in Table 12, perplexity improves smoothly as the beam width increases, but the model achieves near-optimal results with a width of just $M=4$. This rapid convergence indicates that the exact top-k tail is unnecessary; capturing the core semantic cluster is sufficient for effective modeling.
>
>
> | Beam-width \(M\) | 1     | 2     | 4     | 8     |
> |------------------|-------|-------|-------|-------|
> | Perplexity ↓     | 20.7  | 19.4  | **18.5** | 18.6  |
>
> **Conclusion** LMs are **not** exact K-NN systems; they benefit from semantic redundancy and the "hydra effect" [2] of multi-head attention, which compensate for retrieval approximations. By matching standard ANN algorithms in recall while being 6–100× faster, HiRouter provides the optimal balance for Transformer attention.
>
> [1] Big Bird: Transformers for Longer Sequences [2] The Hydra Effect: Emergent Self-repair in Language Model Computations
>
> ### W4-3: Which piece yields non-trivial gains
>
> We emphasize that a critical advantage of HiRouter over classical indexing is that **attention keys and queries are learnable**. Unlike static databases, our approach allows the routing loss to actively regularize representations, forcing them to form natural hierarchical clusters. This joint optimization aligns the data manifold with the router's tree structure, making top-k retrieval significantly more effective than applying a router to fixed vectors.
>
> Beyond improved retrieval recall, this representation learning offers two distinct benefits:
>
>
> 1. **Sharper Clustering:** The dual-entropy loss minimizes assignment uncertainty, producing high-quality, cluster-structured representations.
>
> 2. **Hierarchical Semantic Priors:** The induced hierarchy captures coarse-to-fine semantic groupings, effectively injecting structural priors that improve downstream language modeling performance.
>
> **Empirical Evidence: Training vs. Non-Training** To quantify the impact of this joint optimization, we compare the recall of our trained HIROUTER against a baseline where the router is applied to untrained representations (`num_levels=4`, `num_buckets=4`, and `seq_length=4096`).
>
>
> Beam Width (M)| 8 | 12 | 16 | 32 | 64
> -|-|-|-|-|-|
> Trained / Recall @128 | 25.3% | 34.0% |39.4% | 62.1% | 83.0%
> Without Train / Recall @128| 8.8%|  13.1% | 15.6% | 27.0% | 46.1% |
>
> **Conclusion:** The results above show a **2x–3x improvement** in recall when the router and representations are co-trained. This confirms that our performance gains are driven by the specific synergy between the Hierarchical Router and our dual-entropy loss design, rather than the router architecture alone.
>
>
> ---
>
> In summary, we would like to express once again our sincere gratitude for the reviewer’s time, thoughtful feedback, and valuable suggestions, which have greatly helped us improve the quality and clarity of this work. We appreciate the opportunity to address the comments and are happy to engage in any further discussion if possible.

---

### Author Response · Authors · 2025-12-03
**Post-Rebuttal Discussion Summary**

**Discussion Status:**

Reviewers `cdKJ` and `jZmf` confirmed that **concerns about clarity, scalability, and latency were addressed** and maintained their positive assessments. Reviewer `MiJc` **did  not participate in the discussion**, but our added Recall@k results, expanded baselines,
and clearer distinctions from traditional retrieval methods directly address the issues they raised.


**Comment:**
We thank the ICLR Reviewers, ACs, and PCs for their thoughtful engagement with our submission. Given the shortened discussion period, we provide a concise summary of our contributions, the reviewers’ assessments, and the revisions and clarifications added during the rebuttal period.

**Our contributions:**
In this work, we introduce **HiRouter**, a hardware-efficient sparse attention mechanism that achieves **O(T)** complexity via hierarchical routing. Unlike static clustering or cache-compression approaches, HiRouter **jointly optimizes** the router and Q/K representations using a **dual-entropy objective** that stabilizes training and enforces balanced routing for GPU parallelism. HiRouter achieves **state-of-the-art perplexity among sparse attention methods**, outperforms Routing Transformer and NSA, and **matches or exceeds FlashAttention speed** at long sequence lengths, while remaining end-to-end trainable and scalable to large models.

Below is the summary of reviewers’ scores and impressions:

- **Reviewer cdKJ:** 6 (Positive overall; explicitly noted improved impression post-rebuttal.)
- **Reviewer jZmf:** 6 (Positive; satisfied with algorithmic clarification and complexity analysis.)
- **Reviewer MiJc:** 2 (Did not participate in the discussion; concerns are regarding baselines and novelty, and we have now provided clear baseline comparisons, additional baseline experiments, and concise methodological novelty clarifications that directly address these points.)

Main clarifications and revisions made during the rebuttal:

- **Differentiation from prior work:** We clarified that SubGen and ClusterKV are cache-compression techniques operating on fixed embeddings, not modifications to attention mechanisms. We emphasized that ZETA relies on very low-dimensional projections and is incompatible with Rotary Position Embedding (ROPE), explaining its degradation and why HiRouter’s jointly-trained router achieves substantially higher recall and lower perplexity.
- **Scalability and large-scale evaluation:** We added **new 1B-parameter experiments** trained on **10B and 65B tokens**, showing that HiRouter performs comparably to dense Transformers and consistently improves training stability and perplexity across scales.
- **Latency and kernel optimization:** During rebuttal, we updated Triton kernels that remove unnecessary atomics, add full GQA support, and use improved launch configurations. With these updates, HiRouter now achieves **FlashAttention-level latency**, and is **faster for short sequences (4k–8k)**, eliminating the need for hybrid switching.
- **Algorithmic clarity:** We rewrote the routing explanation, added a step-by-step algorithm table, and clarified that sorting is a **single O(T) global operation per level**, preserving linear end-to-end complexity.
- **Experimental transparency:** We incorporated additional Recall@k analyses, qualitative retrieval examples, success-rate comparisons against top-k retrieval baselines, and an ablation with/without QK co-training, clearly demonstrating that joint optimization is essential for effective routing.

**Summary:**
Two reviewers, `cdKJ`, `jZmf` confirmed that their main concerns regarding scale, latency, and clarity were fully addressed during the rebuttal and maintained positive assessments. Although `MiJc` did not join the discussion, we believe our clarifications, especially the expanded baselines, theoretical explanations, and recall@k analyses, directly address the concerns about positioning, indexing novelty, and retrieval effectiveness.
We hope this summary and the substantially strengthened manuscript support a positive recommendation. We sincerely thank the reviewers and AC for their time, constructive feedback, and careful consideration.

---

### Meta-Review · Area_Chair_6z7P · 2026-01-08

**Summary:**

HiRouter proposes a hierarchical routing mechanism for sparse attention. Reviewers raised many concerns on the initial submission, mainly around limited novelty - differentiating proposed approach from huge number of existing works, complexity of the approach and the scaling behavior. Authors presented additional results and comparisons in the response.  Going over the response I agree with reviewer MiJc that there is not substantial novelty in the approach of the paper compared to existing literature. Theoretical results in the paper are trivial and doesn't add any new insights into Transformers or the proposed approach. Overall this is a borderline paper and I recommend rejection.

**Reviewer Concerns:**

Reviewer concerns were mainly around computation cost of the proposed approach, comparison to other clustering style approaches and limited experiments. Authors addressed some of these points in their response.

**Reviewer Scores:**

MiJc  2-> 4.

---

### Decision · Program_Chairs · 2026-01-26

Reject